# GLI3 regulates muscle stem cell entry into $G_{Alert}$ and self-renewal

Caroline E. Brun [ID] [1,2], Marie-Claude Sincennes[1,2], Alexander Y. T. Lin [ID] [1,2], Derek Hall[1,2], William Jarassier[3], Peter Feige[1,2], Fabien Le Grand[3] & Michael A. Rudnicki [ID] [1,2,4 ✉]

Satellite cells are required for the growth, maintenance, and regeneration of skeletal muscle. Quiescent satellite cells possess a primary cilium, a structure that regulates the processing of the GLI family of transcription factors. Here we find that GLI3 processing by the primary cilium plays a critical role for satellite cell function. GLI3 is required to maintain satellite cells in a $G_O$ dormant state. Strikingly, satellite cells lacking GLI3 enter the $G_{Alert}$ state in the absence of injury. Furthermore, GLI3 depletion stimulates expansion of the stem cell pool. As a result, satellite cells lacking GLI3 display rapid cell-cycle entry, increased proliferation and augmented self-renewal, and markedly enhanced regenerative capacity. At the molecular level, we establish that the loss of GLI3 induces mTORC1 signaling activation. Therefore, our results provide a mechanism by which GLI3 controls mTORC1 signaling, consequently regulating muscle stem cell activation and fate.

[1] Sprott Centre for Stem Cell Research, Regenerative Medicine Program, Ottawa Hospital Research Institute, Ottawa, ON K1H 8L6, Canada. [2] Department of Cellular and Molecular Medicine, Faculty of Medicine, University of Ottawa, Ottawa, ON K1H 8M5, Canada. [3] Univ Lyon, Univ Lyon 1, CNRS, INSERM, Pathophysiology and Genetics of Neuron and Muscle, UMR5261, U1315, Institut NeuroMyoGène, 69008 Lyon, France. [4] Department of Medicine, Faculty of Medicine, University of Ottawa, Ottawa, ON K1H 8M5, Canada. ✉email: mrudnicki@ohri.ca

Adult muscle stem cells, or satellite cells, reside within their niche between the basal lamina and myofiber sarcolemma in a reversible $G_0$ quiescent state[1]. Upon muscle damage, the niche is modified, inducing the transition from quiescence to activation in satellite cells. This process triggers mechano-property changes, migration, metabolic activation, increased RNA transcription and protein synthesis, and cell cycle entry[2–6]. Once activated, satellite cells proliferate extensively to generate myogenic progenitors that differentiate and fuse to repair the injured myofibers, while a subset of satellite cells self-renew and return to quiescence to replenish the stem cell pool[1]. Hence, the ability to self-renew and reversibly enter quiescence is a hallmark of satellite cells that ensures proper muscle repair throughout life. Interestingly, in response to extrinsic cues, satellite cells can dynamically transit from a $G_0$ to a $G_{Alert}$ quiescent state, which is mediated by mTORC1 signaling[6–9]. Whether other pathways regulate this poised activation state that confers enhanced regenerative capacity to the $G_{Alert}$ satellite cells remains unestablished.

A high proportion of quiescent satellite cells harbor a primary cilium, which rapidly disassembles upon activation and reassembles preferentially in self-renewing satellite cells[10]. The primary cilium is a small, non-motile, microtubule-based structure anchored to a cytoplasmic basal body that protrudes from cells in $G_0$[11]. It acts as a nexus for cellular signaling, most notably Hedgehog signaling[12,13], wherein cilia-mediated processing of the GLI family of transcription factors is required for Hedgehog signal transduction[14–16]. Although there are three GLI transcription factors (GLI1-3), only GLI3 contains a potent N-terminal repressor domain.

In the absence of Hedgehog ligands, GLI3 is sequentially phosphorylated at the base of the cilium, first by the cAMP-dependent protein kinase A (PKA) and then by glycogen synthase kinase 3 (GSK3) and casein kinase 1 (CK1)[17–20]. GLI3 phosphorylation promotes its proteolytic cleavage, converting it to a repressor form. Accordingly, decreased ciliary PKA activity leads to the activation of Hedgehog signaling independently of Hedgehog ligand-receptor binding[21–24]. Hedgehog ligand-receptor binding induces the accumulation of GLI3 at the cilium tip, thereby limiting its phosphorylation and cleavage[14,19]. Thus, the primary cilium controls the balance between the full-length activator (GLI3FL) and cleaved repressor (GLI3R), which typically dictates the activation state of Hedgehog signaling.

Although studies have demonstrated that primary cilia act as a signaling hub, their role in satellite cell function remains unknown. Here, we identify GLI3 as a mediator of the cell-autonomous, cilia-related control of satellite cell function. Using a conditional knockout strategy, we show that the repressor form of GLI3 (GLI3R) controls the expansion and self-renewal of muscle stem cells through the regulation of mTORC1 signaling. Moreover, we find that the genetic ablation of *Gli3* promotes $G_0$ to $G_{Alert}$ transition of quiescent muscle stem cells. Overall, the $G_{Alert}$ transition and the increased ability to self-renew confer the enhanced regenerative capacity to the *Gli3*-depleted satellite cells. We, therefore, suggest that primary cilia-mediated GLI3 processing controls the activation and regenerative ability of muscle stem cells.

## Results

***Gli3* displays high expression in satellite cells**. Satellite cells harbor a primary cilium[10,25], yet the cilia-mediated pathways regulating satellite cell function remain uncharacterized. As the Hedgehog pathway requires the primary cilium for its transduction[12], we hypothesized that cilia-mediated Hedgehog signaling regulates satellite cell function.

We first performed RNA-sequencing on freshly isolated satellite cells from resting and injured muscles. Although the FACS method induces partial activation[26–28], we refer to the freshly isolated satellite cells (SCs) as quiescent satellite cells (QSCs) in the manuscript. In addition, the mixed population of proliferating satellite cells and progenitors isolated from 3 days post-cardiotoxin (CTX)-injured muscles will be referred as to activated satellite cells (ASCs)[29]. We observed that the components of canonical Hedgehog signaling, namely the receptor *Patched1* (*Ptch1*), the signal transducer *Smoothened* (*Smo*), and the three transcriptional effectors *Gli1*, *Gli2*, and *Gli3*, are expressed in QSCs (Supplementary Fig. 1a). However, expression of conserved GLI-target genes[30], such as *Gli1*, *Ptch1*, *Ptch2*, *Hhip*, and *Bcl2*, is significantly downregulated as satellite cells transit from quiescence to activation (Supplementary Data file 1 and Supplementary Fig. 1b). The increased expression of *Cdk6*, *Ccnd2*, and *Ccnd1* is likely due to the cycling state of the ASCs.

Interestingly, the transcriptional effector *Gli3* is enriched in ASCs, while the expression of *Gli1* and *Gli2* is strikingly downregulated (Supplementary Data file 1 and Supplementary Fig. 1b). Expression analyses on QSCs, ASCs, proliferating myoblasts, and differentiated myotubes further confirmed that only *Gli3* is enriched in ASCs and proliferating myoblasts (Fig. 1a), supporting the hypothesis that GLI3 is a key player in Hedgehog signal transduction across myogenic progression.

**Cilia dynamics controls GLI3 processing in myogenic cells**. As GLI3 activity relies on the primary cilium[15,19], we investigated the kinetics of primary cilia assembly and disassembly on cultured myofibers and myoblasts (Supplementary Fig. 1c). Primary cilia were labeled by immunostaining using antibodies directed against the primary cilium-specific protein ARL13B and acetylated α-TUBULIN (acαTUB) (Fig. 1). Only acαTUB⁺/ARL13B⁺ elongated structures (>0.5 μm) were enumerated to avoid counting ciliary vesicles as cilia. Hence, on freshly isolated myofibers from the extensor digitorum longus (EDL) muscle, we found that more than 60% of the satellite cells have a primary cilium with an average length of 1.44 μm (Fig. 1b–d). The length was slightly reduced compared to previous reports[10]. This may be due to our method and time of digesting single myofibers, which results in partial activation of muscle stem cells[26–28]. When satellite cells enter the cell cycle (Fig. 1b, Activation), primary cilia length decreases sharply. Then, primary cilia disassemble prior to mitosis (Fig. 1b, Proliferation) to reassemble specifically in the PAX7⁺ self-renewing cells[10] (Fig. 1b, Self-renewal). Accordingly, less than 10% of proliferating myoblasts exhibit a primary cilium longer than 0.5 μm and upon differentiation induced by serum starvation, elongated primary cilia reappear in the PAX7⁺ 'reserve' cells[31] specifically (Fig. 1c–e).

Full-length and non-PKA phosphorylated GLI3 (GLI3FL) localizes throughout the primary cilium, while the cleaved repressor GLI3R is found at the basal body of the cilium where its proteolytic processing occurs[14,19,32]. To assess the temporal coordination of muscle cell ciliation and both GLI3 localization and proteolytic processing, we performed immunostaining and Western blot on satellite cells and primary myoblasts (Fig. 2 and Supplementary Fig. 2). In cycling satellite cells and myoblasts devoid of primary cilia, GLI3 localizes mainly in the cytoplasm and around the microtubule-organizing centers (Fig. 2a, T0 and Supplementary Fig. 2a, b). However, when they exhibit primary cilia, 65% of them display GLI3 accumulation at the axoneme and the tip of primary cilia (Fig. 2a, T0). Accordingly, the GLI3FL protein level is the highest in proliferating myoblasts (Fig. 2c–f, T0). When myoblasts undergo differentiation, GLI3 progressively transits from the axoneme to the basal body of primary cilia

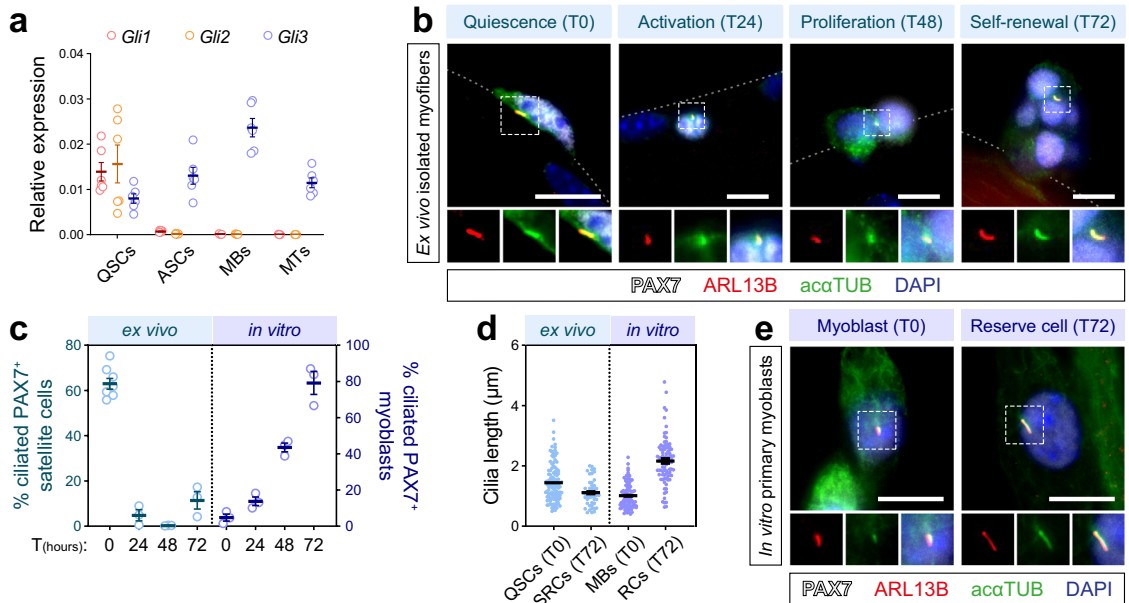

**Fig. 1 Primary cilia are dynamically regulated in muscle cells. a** *Gli1*, *Gli2*, and *Gli3* expression determined by RT-qPCR and normalized to *Ppia* and *Rps18* in QSCs, ASCs, myoblasts (MBs), and 3 days-differentiated muscle cells or myotubes (MTs) ($n = 6$ (three males and three females for each genotype)). **b** Representative immunofluorescence staining of PAX7 (white), ARL13B (red), acetylated alpha-TUBULIN (acαTUB, green), and nuclei (blue) showing primary cilia on freshly isolated (T0) and cultured myofibers (T24-72). **c** [ex vivo] Proportion of PAX7+ quiescent (T0), activated (T24), proliferating (T48), and self-renewing (T72) satellite cells harboring a primary cilium (ARL13B+/acαTUB+, > 0.5 µm) on myofibers ($n = 8$ mice for T0; $n = 3$ mice for T24, T48, and T72). [in vitro] Proportion of PAX7+ primary myoblasts harboring a primary cilium during the 72-h-differentiation time course ($n = 3$ biologically independent samples). **d** Cilia length measurement on quiescent (QSCs) and self-renewing (SRCs) PAX7+ satellite cells, PAX7+ myoblasts (MBs), and reserve cells (RCs) (QSCs, $n = 112$ and SRCs, $n = 48$ cells examined from three mice; MBs, $n = 97$ and RCs, $n = 48$ cells examined from six biologically independent samples). **e** Representative immunostaining of PAX7 (white), ARL13B (red), acαTUB (green), and nuclei (blue) showing primary cilia on a myoblast (MB) (T0) and a reserve cell (RC) (T72). Scale bars, 10 µm. Means ± SEM.

(Fig. 2a, b, T24-T72), concomitantly increasing the level of GLI3R at the expense of GLI3FL (Fig. 2c–f). At 72 h, the PAX7+ 'reserve' cells have a cilium and maintain a high GLI3R level (Fig. 2, T72, RCs), whereas differentiated myotubes have no primary cilium and do not express GLI3 (Fig. 2c, d, T72, MTs). As observed in the reserve cells, no GLI3 staining was detected at the primary cilium of QSCs, except for a few cells in which the GLI3 antibody clearly labels the basal body, where PKA is expressed (Supplementary Fig. 2a, b). Although reserve cells are not bona fide QSCs, they display a similar ciliation profile, suggesting GLI3 is mainly processed as a repressor in QSCs.

To confirm that cilia- and PKA-mediated regulation of GLI3 processing is conserved in muscle cells, primary myoblasts were treated with either forskolin (FSK), an adenylyl cyclase activator that stimulates PKA[19,32], as a mean to increase GLI3 processing, or with siRNA against *Ift88* to disrupt primary cilia assembly and GLI3 processing[16]. Additionally, myoblasts were treated with SAG, a SMOOTHENED agonist that induces accumulation of GLI3 in the primary cilium, thus abrogating its proteolytic processing[19,33]. Accordingly, SAG-treated myoblasts exhibit GLI3 ciliary accumulation, while FSK treatment prevents it and promotes GLI3 proteolytic cleavage (Supplementary Fig. 2c–g). Knocking-down *Ift88* decreases GLI3R levels, consequently increasing GLI3FL/GLI3R ratio (Supplementary Fig. 2h–j). Hence, FSK treatment decreases the expression of the two well-known GLI-target genes, *Gli1* and *Ptch1*, whereas *siIft88* and SAG treatments increase it (Supplementary Fig. 2k–n). Importantly, *Gli3* siRNA treatment induces similar upregulation of *Gli1* and *Ptch1* (Supplementary Fig. 2m), indicating that GLI3FL is dispensable for GLI-target gene expression and that the predominant regulatory form is the GLI3R repressor. These results indicate that GLI3 processing regulated by ciliary PKA is

conserved in myoblasts and suggest that a similar process occurs in satellite cells.

Collectively, these results show that transient primary cilia disassembly during cell cycle entry and proliferation abolishes GLI3R processing, promoting ciliary GLI3FL accumulation. Upon differentiation, myoblasts reassemble a primary cilium that will be maintained in the self-renewing "reserve" cell population only. GLI3 leaves the primary cilium and is mainly expressed as a repressor. Thus, our results suggest that the ratio of GLI3FL/GLI3R relies, at least in part, on cilia dynamics, as satellite cells progress through the myogenic lineage and self-renew.

**GLI3R controls the regenerative potential of satellite cells**. To characterize the role of GLI3 in regulating satellite cell function, we generated *Pax7*CE/+; *Gli3*+/+; *R26R*YFP and *Pax7*CE/+; *Gli3*fl/fl; *R26R*YFP (hereafter referred to as *Gli3*+/+ and *Gli3*Δ/Δ mice, respectively), in which *Gli3* was excised in satellite cells that are simultaneously labeled by YFP upon tamoxifen induction (Supplementary Fig. 3a). The efficiency of *Gli3* excision was confirmed in QSCs and their myogenic progeny (ASCs, MBs, and MTs) (Supplementary Fig. 3b, c).

Uninjured resting muscles exhibit no gross histological abnormalities upon tamoxifen treatment, although *Gli3*Δ/Δ muscles display a slight increase in satellite cell number (Supplementary Fig. 3d, e). This observation led us to hypothesize that *Gli3* deletion in satellite cells could be inducing a break from quiescence. During homeostasis, satellite cells reside in their niche, which maintains their quiescence, but then migrates outside the basal lamina upon activation[2,5]. Analyzing muscle cross-sections post-tamoxifen injection showed that both *Gli3*+/+ and *Gli3*Δ/Δ satellite cells are located in their niche, underneath

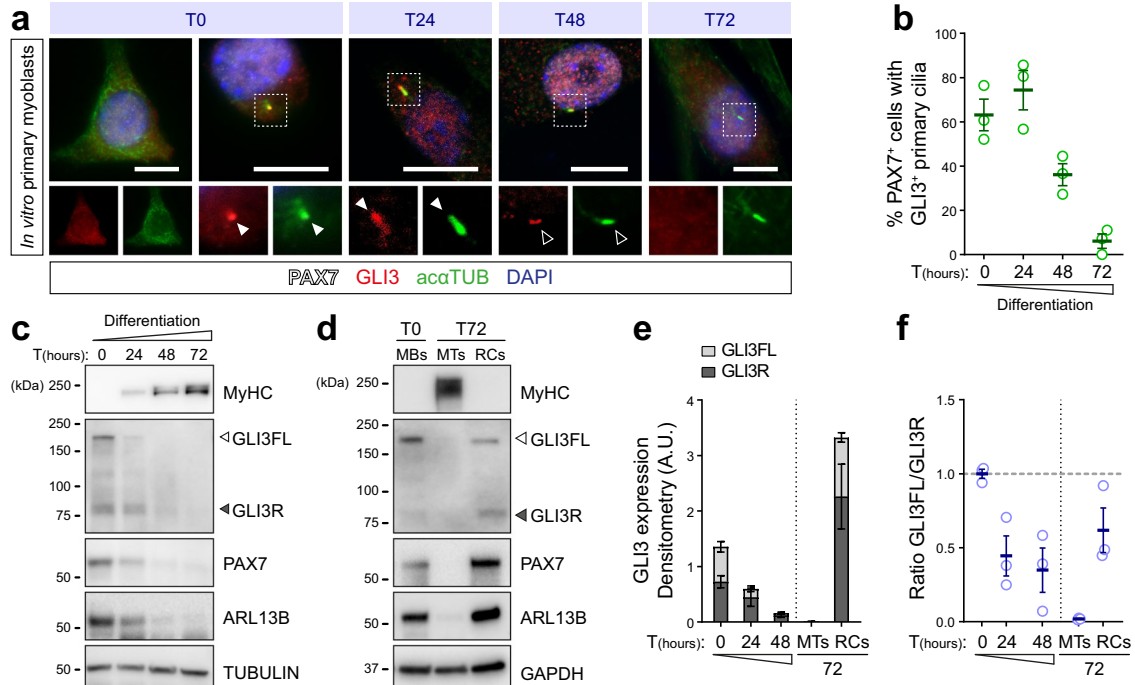

**Fig. 2 GLI3 subcellular localization and proteolytic processing rely on primary cilia dynamics. a** Immunostaining of GLI3 (red) in primary cilia labeled by acetylated α-TUBULIN (acαTUB, green) of PAX7+ primary myoblasts (white) during their differentiation time course. White arrows indicate the tip of the primary cilium; empty arrows indicate the basal body. **b** Proportion of PAX7+ cells that display a GLI3+ primary cilium during myoblast differentiation time course ($n = 3$ biologically independent samples). **c** Immunoblot analysis of GLI3 full-length (GLI3FL) and repressor (GLI3R) and the ciliary membrane marker ARL13B in proliferating myoblasts from 0 to 72 h of differentiation. PAX7 and MyHC (myosin heavy chain) are used to monitor myogenic differentiation. TUBULIN is used as a loading control. **d** Immunoblot analysis of ARL13B, GLI3FL, and GLI3R in myoblasts (MBs), 72 h-differentiated myoblasts or myotubes (MTs), and reserve cells (RCs). Both MBs and RCs express PAX7 while MTs express the myogenic differentiation marker MyHC. GAPDH is used as a loading control. **e** Densitometric analysis of the level of GLI3FL and GLI3R relative to TUBULIN signals of six (−24 h) and three biological replicates (0–72 h). **f** Ratio of GLI3FL/GLI3R relative to GAPDH (−24 h, $n = 6$ biological replicates; 0–72 h, $n = 3$ biological replicates). Scale bar, 10 μm. Means ± SEM.

the basal lamina (Supplementary Fig. 3f, g). Together, these results indicate that $Gli3^{\Delta/\Delta}$ satellite cells are inherently in a quiescent state and suggest that Cre-mediated $Gli3$ deletion may favor transient activation, slightly increasing their number over weeks after tamoxifen treatment.

To assess their regenerative potential, $Gli3^{+/+}$ and $Gli3^{\Delta/\Delta}$ mice were subjected to a CTX-induced injury in the *tibialis anterior* (TA) muscle (Fig. 3a). At 7 days post-injury (d.p.i.), we counted the number of self-renewing and differentiating cells expressing either PAX7 or MYOGENIN (MYOG), respectively. An increased number of PAX7+ SCs was observed in $Gli3^{\Delta/\Delta}$ mice (Fig. 3b, c), while the number of MYOG+ cells is slightly decreased compared to $Gli3^{+/+}$ mice (Supplementary Fig. 3h, i). The morphology of the nascent myofibers expressing DYSTRO-PHIN (DYS) is improved in $Gli3^{\Delta/\Delta}$ mice, where interstitial space is reduced and myofibers are larger, suggesting that the delay in a myogenic commitment does not impact the overall efficiency of regeneration (Fig. 3b and Supplementary Fig. 3h). The number of PAX7+ SCs is also increased in $Gli3^{\Delta/\Delta}$ mice at 21 d.p.i., suggesting enhanced self-renewal and expansion of the satellite cell pool (Fig. 3d, e). Additionally, there is a significant increase in muscle mass mainly due to myofiber hypertrophy (Fig. 3f, g and Supplementary Fig. 3j, k) and increased number of centrally located nuclei (Fig. 3h). This increase in myonuclear accretion suggests that, although there is a delay in terminal differentiation, the initial expansion of the satellite cell pool ultimately leads to a higher number of myogenic progenitors that eventually fuse into nascent and regenerating myofibers. Finally, in situ measurements of muscle force revealed that $Gli3^{\Delta/\Delta}$ regenerated TA

muscles are ~50% stronger than $Gli3^{+/+}$ muscles (Fig. 3i and Supplementary Fig. 3l), showing that $Gli3$ deletion in satellite cells improves muscle regeneration after CTX-induced trauma.

We finally assessed the long-term regenerative capacity of $Gli3^{\Delta/\Delta}$ satellite cells by challenging the mice with repetitive muscle injuries (Fig. 3j). Strikingly, $Gli3^{\Delta/\Delta}$ regenerated muscles are bigger, though some regions exhibit clear histological defects with fibrotic areas and tiny myofibers (Supplementary Fig. 3m). Even after three rounds of injury, $Gli3^{\Delta/\Delta}$ mice maintain a high number of self-renewing satellite cells (Fig. 3k, l). As observed following a single injury, $Gli3^{\Delta/\Delta}$ mice exhibit increased muscle weight resulting from both myofiber hyperplasia and hypertrophy (Fig. 3m and Supplementary Fig. 3n–p), and regenerating myofibers contain a higher number of centrally located nuclei (Supplementary Fig. 3q).

Altogether, our data suggest that GLI3 acts at multiple levels regulate satellite cell function, controlling their quiescence, expansion, and regenerative capacity during each regeneration cycle.

**GLI3R has pleiotropic functions in satellite cells.** To assess more precisely the role of GLI3 in regulating satellite cell function, single EDL myofibers from $Gli3^{+/+}$ and $Gli3^{\Delta/\Delta}$ mice were isolated and analyzed either directly (0 h) or following 24, 48, and 72 h of culture in order to analyze satellite cell activation, proliferation, differentiation, and self-renewal[34] (Fig. 4a and Supplementary Fig. 4a).

Post-isolation, a comparable proportion of ciliated cells were found on freshly isolated myofibers from $Gli3^{+/+}$ and $Gli3^{\Delta/\Delta}$ mice (Fig. 4b, c). Consistent with our in vivo quantification of

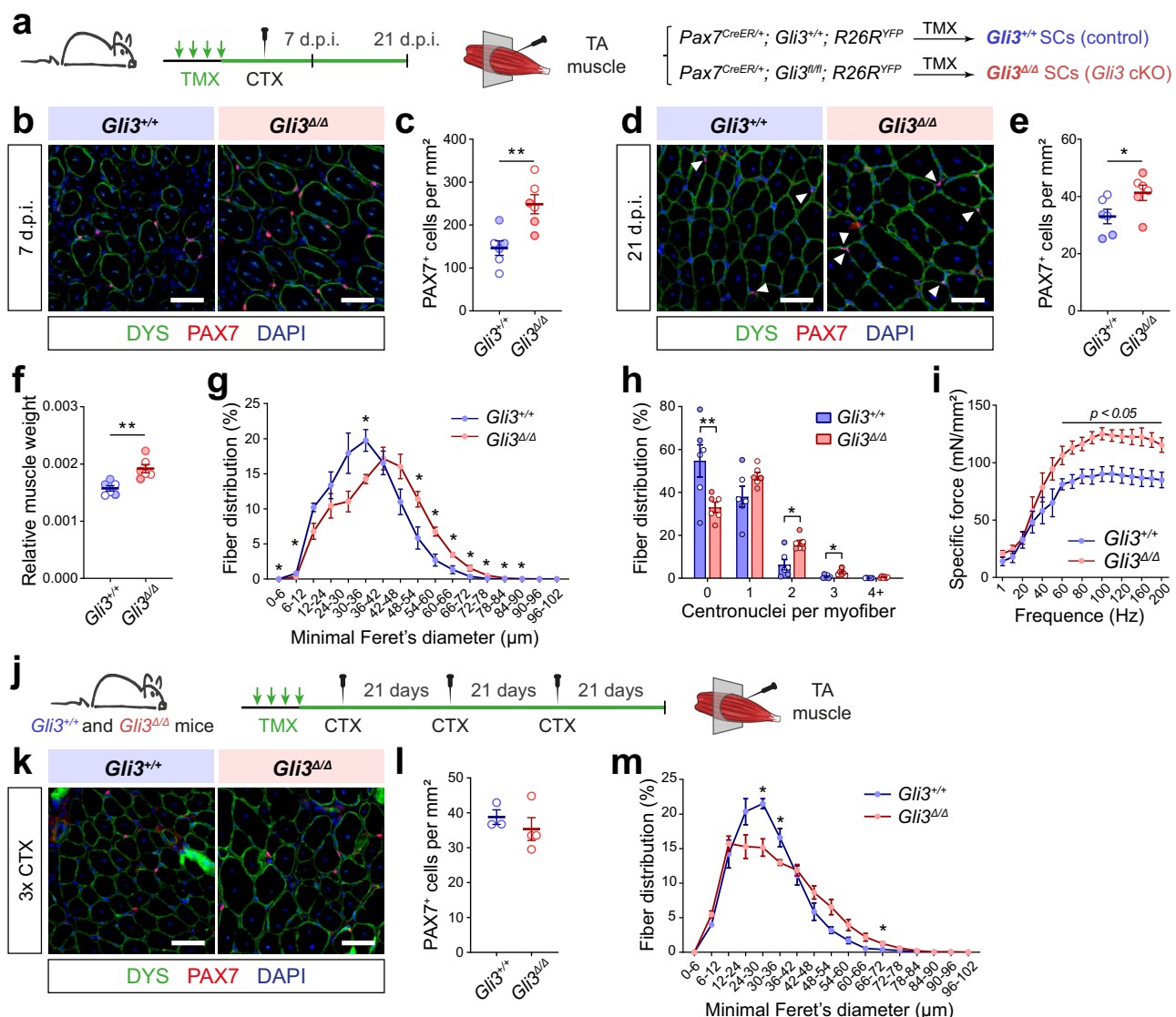

**Fig. 3 Gli3 deletion in satellite cells enhances muscle regeneration. a** Experimental design to analyze cardiotoxin (CTX)-induced muscle regeneration of $Gli3^{+/+}$ and $Gli3^{\Delta/\Delta}$ mice. **b** Representative images of TA muscle sections at 7 days post-injury (d.p.i.). Satellite cells (SCs) are labeled with PAX7 (red). DYSTROPHIN (DYS, green) delineates regenerating myofibers. **c** Quantification of PAX7+ SCs per mm² TA section at 7 d.p.i. **d** Immunostaining of PAX7 (red) and DYSTROPHIN (green) on TA sections at 21 d.p.i. **e** Quantification of PAX7+ SCs per mm² TA section at 21 d.p.i. **f** TA muscle weight normalized to total body weight. **g** TA myofiber distribution according to minimal Feret's diameter. **h** Regenerated TA myofiber distribution according to their number of centrally located nuclei (centronuclei) at 21 d.p.i. **i** Specific force of regenerated TA muscles at 21 d.p.i. **j** Experimental design to analyze regeneration of $Gli3^{+/+}$ and $Gli3^{\Delta/\Delta}$ mice following three consecutive CTX-injuries in the TA muscle (3xCTX). **k** Immunostaining of PAX7 (red) and DYSTROPHIN (green) on TA muscle sections following 3xCTX. **l** Quantification of PAX7+ SCs per mm² 3xCTX injured TA section. **m** Regenerated TA myofiber distribution according to minimal Feret's diameter. DAPI stains nuclei (blue). Scale bars, 50 μm. $n = 6$ (three males, empty dots, and three females, colored dots) for panels **c**, **e**, **f**, **g**, **h**, $n = 5$ males for each genotype for panel **i**, $n = 3$ $Gli3^{+/+}$ males and 4 $Gli3^{\Delta/\Delta}$ males for panels **l**, **m**. Means ± SEM; Unpaired $t$-test with Welch's correction for panels **c**, **e**, **f**, **h**, **l**, Multiple unpaired $t$-test for panels **g**, **i**, **m**, *$p < 0.05$; **$p < 0.01$.

PAX7+ satellite cells in resting muscle (Supplementary Fig. 3d, e), we observed that $Gli3^{\Delta/\Delta}$ myofibers exhibit a higher number of satellite cells (Supplementary Fig. 4b). Interestingly as well, the primary cilia length of $Gli3^{\Delta/\Delta}$ satellite cells is shortened (Fig. 4d), suggesting that $Gli3^{\Delta/\Delta}$ satellite cells may be poised to activate faster. This was confirmed using myofibers cultured with EdU for 24 h, where $Gli3^{\Delta/\Delta}$ myofibers show a higher proportion of EdU+ satellite cells compared to $Gli3^{+/+}$ (Fig. 4e, f), indicating that $Gli3^{\Delta/\Delta}$ satellite cells enter the cell cycle faster. At 48 h, following 1h-EdU pulse, $Gli3^{\Delta/\Delta}$ satellite cells exhibit increased EdU incorporation along with an increase in the number of satellite cells per myofiber (Fig. 4g, h and Supplementary Fig. 4b). Finally, at 72 h, the proportion of MYOG+ cells was decreased upon $Gli3$

deletion (Fig. 4i, j), consistent with increased self-renewal at the expense of differentiation.

Therefore, these data further support the hypothesis that GLI3 has multiple roles from maintaining satellite cell quiescence to regulating satellite cell fate.

**GLI3R regulates G_Alert entry of quiescent satellite cells.** To characterize the consequences of $Gli3$ deletion, we performed RNA-sequencing on $Gli3^{+/+}$ and $Gli3^{\Delta/\Delta}$ QSCs and ASCs and compared their transcriptome (Fig. 5a, b, Supplementary Fig. 5a, and Supplementary Data file 2). Principle component analysis (PCA) of the transcriptional profiles of $Gli3^{+/+}$ and $Gli3^{\Delta/\Delta}$ SCs

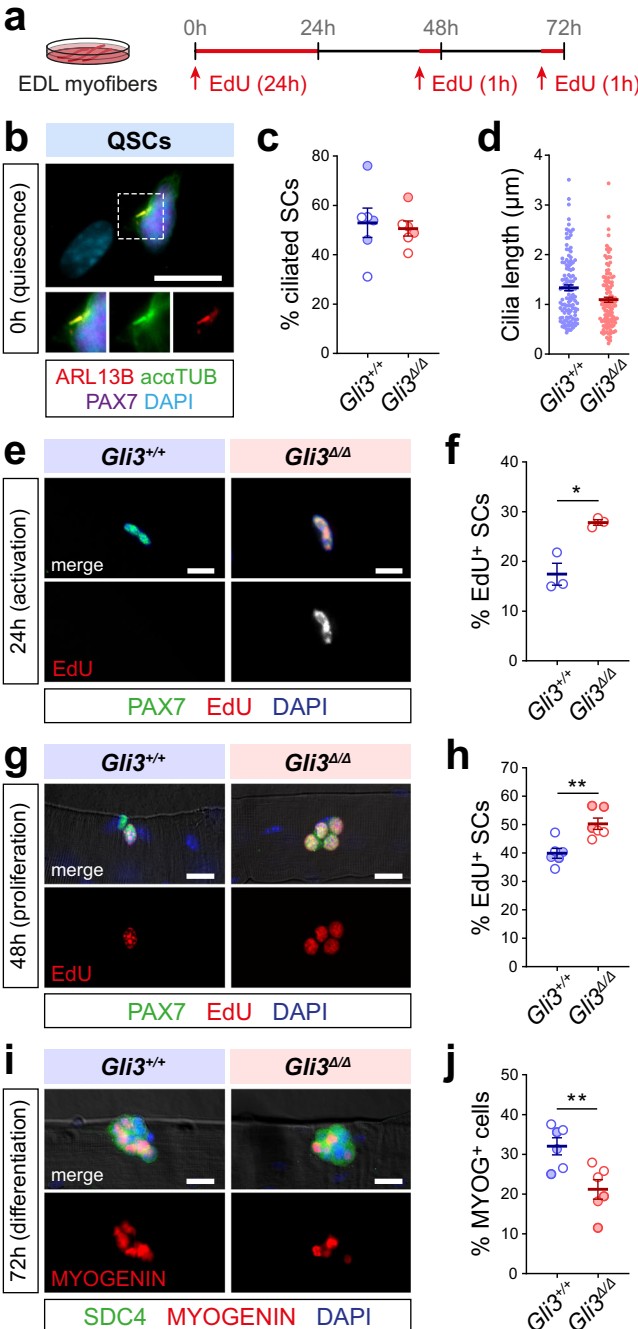

**Fig. 4 GLI3 regulates multiple aspects of satellite cell function.**
**a** Experimental design to study satellite cell activation, proliferation, differentiation, and self-renewal on isolated EDL myofibers. **b** Primary cilia are immunostained with ARL13B (red) and acαTUB (green) and satellite cells with PAX7 (purple). **c** Proportions of PAX7+ ciliated satellite cells on freshly isolated myofibers from $Gli3^{+/+}$ and $Gli3^{\Delta/\Delta}$ mice. **d** Length of primary cilia (ARL13B+/acαTUB+, >0.5 μm) on quiescent satellite cells from freshly isolated $Gli3^{+/+}$ and $Gli3^{\Delta/\Delta}$ myofibers ($n = 3$ $Gli3^{+/+}$ mice, $n = 4$ $Gli3^{\Delta/\Delta}$ mice). **e** Representative immunofluorescence of satellite cells (PAX7, green) having incorporated EdU (red) on 24h-cultured myofibers. **f** Proportions of EdU+ satellite cells 40 h after isolation from $Gli3^{+/+}$ and $Gli3^{\Delta/\Delta}$ mice ($n = 3$ males). **g** Representative immunofluorescence of satellite cells (PAX7, green) having incorporated EdU (red) on 48-h-cultured myofibers after 1 h EdU incubation. **h** Proportion of $Gli3^{+/+}$ and $Gli3^{\Delta/\Delta}$ satellite cells having incorporated EdU. **i** Representative immunofluorescence using anti-SYNDECAN-4 (SDC4, green) to label all satellite cells and MYOGENIN (MYOG, red) to mark the differentiated ones. **j** Quantification of the number of MYOG+ satellite cells per $Gli3^{+/+}$ and $Gli3^{\Delta/\Delta}$ myofiber after 72 h of culture. Nuclei are stained with DAPI (blue). Scale bars, 10 μm. Unless otherwise stated, $n = 6$ (three males, empty dots, and three females, colored dots). Means ± SEM; Unpaired $t$-test with Welch's correction for panels **f**, **h**, **j**, *$p < 0.05$; **$p < 0.01$.

---

transcripts in $Gli3^{\Delta/\Delta}$ QSCs revealed enrichment for biological process terms related to glucose, insulin signaling and oxidative phosphorylation, all intersecting with mTORC1 signaling activity (Fig. 5c and Supplementary Data file 2). mTORC1 signaling activity drives the $G_0$-to-$G_{Alert}$ transition in quiescence[6]. Moreover, since we observed that $Gli3^{\Delta/\Delta}$ mice have an increased number of satellite cells (Supplementary Figs. 3d, e, 4b) and their QSCs display shortened cilia and enter the cell cycle faster (Fig. 4b–f), we hypothesized that $Gli3^{\Delta/\Delta}$ SCs have transitioned into $G_{Alert}$.

In contrast to quiescent $G_0$ satellite cells, $G_{Alert}$ satellite cells display an increase in cell size, transcriptional activity and mitochondrial metabolism, and are poised to activate faster in response to injury[6]. Imaging flow cytometry revealed that $Gli3^{\Delta/\Delta}$ QSCs display an increase in size compared to $Gli3^{+/+}$ QSCs (Supplementary Fig. 5d–f). $Gli3^{\Delta/\Delta}$ QSCs have increased PyroninY staining, consistent with increased transcriptional activity, and higher mitochondrial mass, in line with our RNA-sequencing data showing enrichment in genes related to oxidative phosphorylation (Fig. 5e, f and Supplementary Fig. 5g). None of these features, however, overlaps with the profiles observed in ASCs, confirming that the $Gli3^{\Delta/\Delta}$ QSCs are not fully activated.

In addition, we analyzed the phosphorylation of the ribosomal protein S6 as a marker for mTORC1 activation in freshly isolated myofibers from $Gli3^{+/+}$ and $Gli3^{\Delta/\Delta}$ mice (Fig. 5g). Remarkably, we observed that more than 50% of the $Gli3^{\Delta/\Delta}$ SCs display staining for phospho-S6, while only 20% of $Gli3^{+/+}$ SCs are phospho-S6+. Finally, culturing $Gli3^{+/+}$ and $Gli3^{\Delta/\Delta}$ QSCs immediately after FACS-isolation in the presence of EdU for 40 h confirmed that $Gli3^{\Delta/\Delta}$ QSCs enter the cell cycle faster (Fig. 5h). Hence, these findings demonstrate that $Gli3^{\Delta/\Delta}$ QSCs have transitioned into $G_{Alert}$.

**GLI3R controls satellite cell proliferation and fate choice.**
Similar in silico analysis was performed on ASCs to determine the molecular consequences of loss of GLI3 (Fig. 6a, b and Supplementary Data file 2). Interestingly, GSEA revealed that $Gli3^{\Delta/\Delta}$ SCs display inhibition of the JAK/STAT signaling (Fig. 6b), which knockdown or pharmacological inhibition was shown to favor satellite cell expansion, homing and regenerative capacity, overall improving skeletal muscle repair[35,36]. Thus, the decreased activity of JAK/STAT signaling in $Gli3^{\Delta/\Delta}$ satellite cells is consistent with

revealed two distinct groups based on the first component axis (PC1), distinguishing the QSCs from the ASCs (Fig. 5b). PCA also revealed a high correlation between $Gli3^{+/+}$ and $Gli3^{\Delta/\Delta}$ QSCs, corroborating the idea that the $Gli3^{\Delta/\Delta}$ SCs are predominantly quiescent. Interestingly, the correlation between the QSC and ASC transcriptional signatures was slightly increased in the $Gli3^{\Delta/\Delta}$ background (0.67) than the $Gli3^{+/+}$ (0.66) (Fig. 5b), suggesting that the $Gli3^{\Delta/\Delta}$ QSCs may actually exhibit some features of satellite cell activation.

In silico analysis revealed that $Gli1$, $Ptch1$, and other GLI-target gene expression increases, albeit non-significant, in the absence of GLI3, further suggesting that GLI3 acts mainly as a repressor in QSCs (Supplementary Fig. 5b, c and Supplementary Data file 2). While no specific enrichment for canonical Hedgehog signaling was found (Fig. 5c, d), Gene Ontology (GO) and gene set enrichment analysis (GSEA) of the 41 significantly upregulated

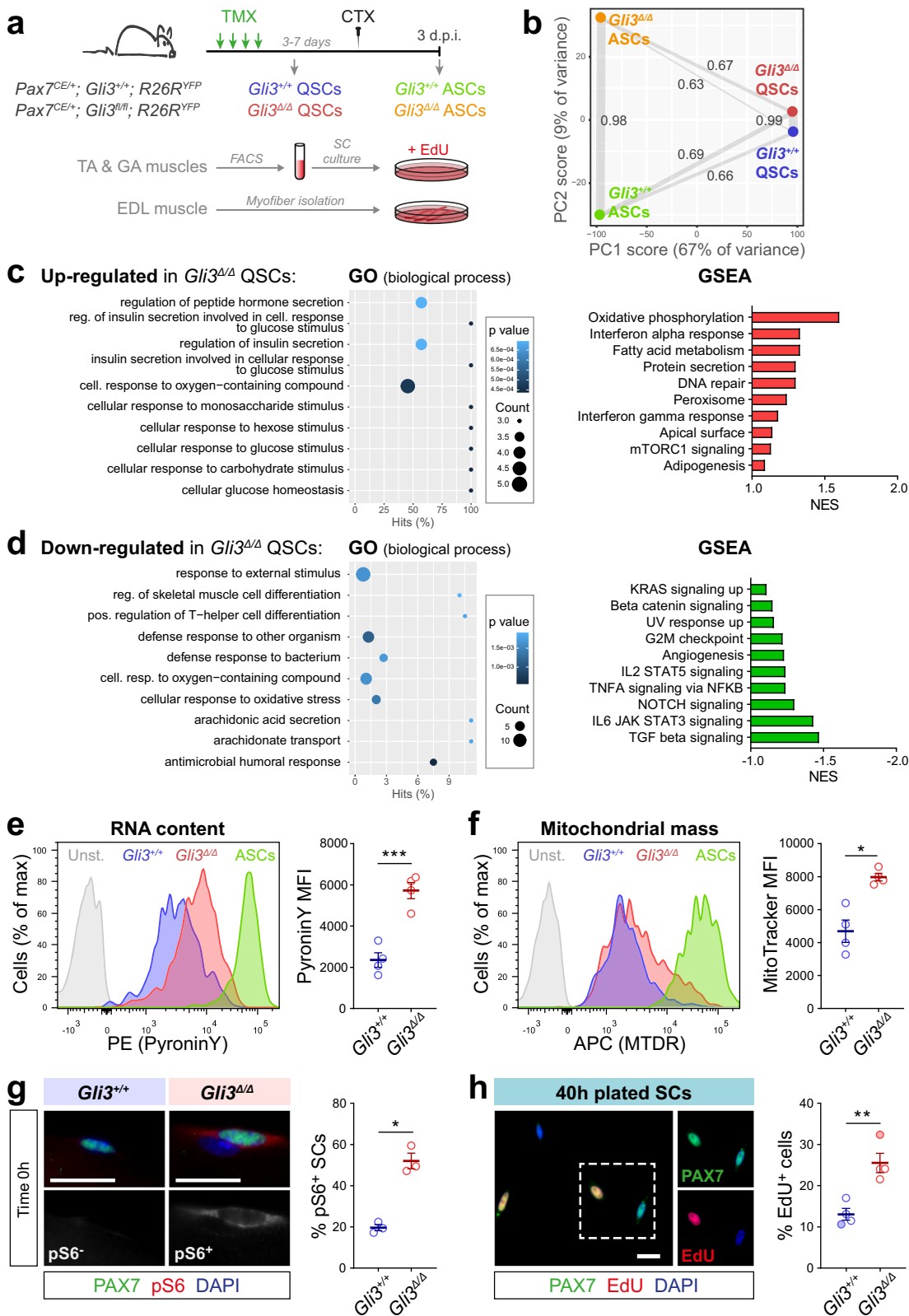

their enhanced ability to self-renew and regenerate. As well, GO analysis and GSEA on the 629 downregulated genes highlighted biological process terms mainly related to cell adhesion, apical junction/surface, and epithelial-mesenchymal transition (Fig. 6b), supporting the idea that $Gli3^{\Delta/\Delta}$ ASCs are less adhesive to their environment during regeneration, allowing more efficient migration and muscle tissue repair[2,37].

In addition, 144 transcripts are significantly upregulated in $Gli3^{\Delta/\Delta}$ ASCs and correlate with GO terms associated with the regulation of G1/S phase transition and cell cycle (Fig. 6a). GSEA further confirmed the activation of cell cycle-related pathways (MYC/E2F targets, G2M checkpoint, DNA repair). Similar to the QSCs, expression of the canonical Hedgehog downstream target genes is not significantly increased in $Gli3^{\Delta/\Delta}$ ASCs

**Fig. 5 Quiescent satellite cells in *Gli3*$^{\Delta/\Delta}$ uninjured muscle are in G$_{Alert}$. a** Experimental design. QSCs and ASCs from TA and *gastrocnemius* (GA) muscles were analyzed for cell size, RNA content, and mitochondrial mass. FACS-isolated SCs were plated 40 h with EdU to analyze cell cycle entry. Phosphorylation of S6 ribosomal protein was analyzed on freshly isolated EDL myofibers. **b** Principal component analysis of global transcriptomes of *Gli3*$^{+/+}$ and *Gli3*$^{\Delta/\Delta}$ QSCs and ASCs and Pearson's values showing a correlation between samples. Each dot represents the mean of three biological samples. **c** Gene ontology (GO) term enrichment and gene set enrichment analysis (GSEA) for the upregulated and **d** downregulated genes in *Gli3*$^{\Delta/\Delta}$ compared to *Gli3*$^{+/+}$ QSCs. **e** Left, Representative plots of *Gli3*$^{+/+}$ and *Gli3*$^{\Delta/\Delta}$ QSCs stained for PyroninY (ASCs activated satellite cells, Unst. unstained). Right, Mean fluorescence intensity (MFI) of PyroninY staining ($n = 4$ males). **f** Left, Representative plots of *Gli3*$^{+/+}$ and *Gli3*$^{\Delta/\Delta}$ QSCs stained for MitoTracker Deep Red (MTDR). Right, MFI of MitoTracker ($n = 4$ males). **g** Left, Immunofluorescence of phospho-S6 (pS6, red) in SCs (PAX7, green) on isolated *Gli3*$^{+/+}$ and *Gli3*$^{\Delta/\Delta}$ myofibers. Right, Proportion of pS6$^+$ SCs ($n = 3$ females). **h** Left, Immunostaining of SCs with PAX7 (green) showing EdU (red) incorporation 40 h post-isolation. Right, Proportion of EdU$^+$ SCs ($n =$ three males, empty dots, and one female, colored dot). DAPI stains nuclei (blue). Scale bars, 10 μm. Means ± SEM; Unpaired *t*-test with Welch's correction for panels **e**, **f**, **g**, **h**, *$p < 0.05$; **$p < 0.01$; ***$p < 0.001$.

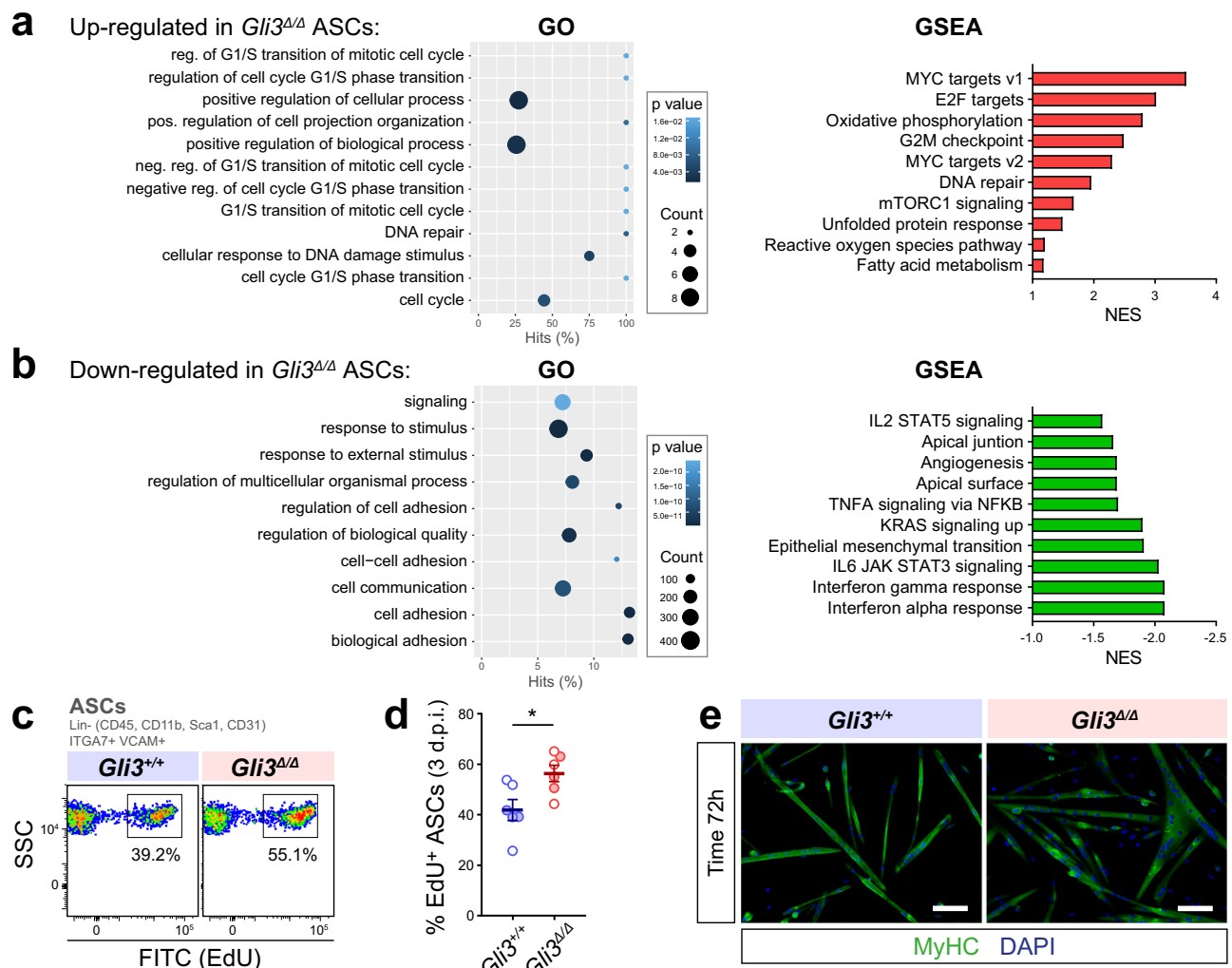

**Fig. 6 *Gli3* deletion allows for the expansion of the satellite cell pool. a** Gene Ontology (GO) term enrichment (Biological process) and gene set enrichment analysis (GSEA) for the upregulated and **b** downregulated genes in *Gli3*$^{\Delta/\Delta}$ ASCs compared to *Gli3*$^{+/+}$ ASCs. **c** Representative flow cytometry plot of *Gli3*$^{+/+}$ and *Gli3*$^{\Delta/\Delta}$ ASCs that have incorporated EdU. **d** Proportion of EdU$^+$ *Gli3*$^{+/+}$ and *Gli3*$^{\Delta/\Delta}$ ASCs isolated from injured muscles at 3 days post-injury (d.p.i.) ($n = 3$ males, empty dots, and three females, colored dots). Means ± SEM; Unpaired *t*-test with Welch's correction, *$p < 0.05$. **e** Representative immunostaining of myosin heavy chains (MyHC, green) of 72-h-differentiated *Gli3*$^{\Delta/\Delta}$ and *Gli3*$^{+/+}$ myoblasts. DAPI (blue) stains nuclei. Scale bar, 100 μm.

(Supplementary Fig. 6b). However, GSEA revealed mTORC1-signaling among the activated pathways in *Gli3*$^{\Delta/\Delta}$ ASCs (Fig. 6a), which promotes satellite cell proliferation, differentiation, and fusion upon activation[38–40].

We previously showed that *Gli3*$^{\Delta/\Delta}$ satellite cells proliferate faster than *Gli3*$^{+/+}$ on cultured myofibers (Fig. 4g, h). To analyze satellite cell proliferation in vivo, *Gli3*$^{+/+}$ and *Gli3*$^{\Delta/\Delta}$ mice were

subjected to a CTX-induced injury in the TA and gastrocnemius muscle and sacrificed 3 days post-injury following a 3h-EdU pulse. Analyzing in vivo EdU incorporation confirmed the enhanced ability of *Gli3*$^{\Delta/\Delta}$ ASCs to proliferate (Fig. 6c, d). Nevertheless, sustained cell proliferation can compromise myogenic differentiation and impair muscle regeneration[36]. To determine whether *Gli3* deletion permanently impairs or

transiently delays myogenic differentiation, we derived primary myoblasts from $Gli3^{+/+}$ and $Gli3^{\Delta/\Delta}$ satellite cells in vitro. $Gli3^{\Delta/\Delta}$ myoblasts exhibit decreased expression of the myogenic regulatory factors, MYOD1, MYOGENIN, and the Myosin Heavy Chain (MyHC), at the early steps of differentiation (0-24 h), yet they express similar levels of MyHC at the later steps (48–72 h) (Supplementary Fig. 6c, d) and even form bigger myotubes than controls (Fig. 6e). This is consistent with the hypertrophy and increased myonuclear accretion we observed in $Gli3^{\Delta/\Delta}$ regenerated muscles (Fig. 3) and suggests that $Gli3^{\Delta/\Delta}$ myoblast differentiation is delayed rather than impaired. Of note, we observed that $Gli3^{\Delta/\Delta}$ myoblasts display increased expression of $Gli1$ and $Ptch1$ (Supplementary Fig. 6c). This is consistent with our results on wild-type myoblasts where the modulation of GLI3R level directly affects $Gli1$ and $Ptch1$ expression (Supplementary Fig. 2). However, in $Gli3^{\Delta/\Delta}$ myoblasts, SAG and FSK treatments failed to change the two GLI-target gene expressions (Supplementary Fig. 6d–f), indicating that GLI1 and GLI2 do not compensate for the absence of GLI3.

Together, these results show that $Gli3$ deletion increases satellite cell proliferation at the expense of early differentiation and delays, but does not prevent, terminal differentiation and fusion of myogenic progenitors.

**GLI3R regulates mTORC1 signaling in muscle cells**. Overall, our data indicate that loss of GLI3 induces $G_0$ QSC transition to $G_{Alert}$ and promotes ASC proliferation, which can all relate to mTORC1 signaling activation. To determine whether GLI3 controls $G_0$ quiescence through a mTORC1-dependent mechanism, we first treated $Gli3^{+/+}$ and $Gli3^{\Delta/\Delta}$ mice with rapamycin (RAPA), a potent inhibitor of mTORC1 signaling (Fig. 7a). RAPA treatment significantly restores $Gli3^{\Delta/\Delta}$ QSC transcriptional activity (PyroninY) and mitochondrial mass (MTDR) to the levels observed in $Gli3^{+/+}$ mice (Fig. 7b, c). This suggests that inhibiting mTORC1 signaling in GLI3-depleted satellite cells blocks satellite cell entry into $G_{Alert}$. Additionally, plating RAPA-treated $Gli3^{+/+}$ and $Gli3^{\Delta/\Delta}$ satellite cells immediately after FACS-isolation in the presence of EdU showed that RAPA-treated $Gli3^{\Delta/\Delta}$ cells lose their ability to enter the cell cycle faster (Fig. 7d). While the proportion of phospho-S6$^+$ is increased in $Gli3^{\Delta/\Delta}$ plated cells compared to $Gli3^{+/+}$ cells, RAPA treatment lowers significantly this proportion in both $Gli3^{+/+}$ and $Gli3^{\Delta/\Delta}$ satellite cells (Supplementary Fig. 7a).

To further assess whether GLI3R mediates mTORC1 signaling in our model, $Gli3^{+/+}$ and $Gli3^{\Delta/\Delta}$ myofibers were treated for 24 h upon isolation with either FSK, as a means to increase GLI3 processing and GLI3R activity, or rapamycin (RAPA) to inhibit mTORC1 signaling (Fig. 7e). While FSK treatment has no effect on $Gli3^{\Delta/\Delta}$ SCs, it decreases the proportion of EdU$^+$ cells in $Gli3^{+/+}$ myofibers (Fig. 7f). As well, FSK diminishes the proportion of pS6$^+$ SCs in $Gli3^{+/+}$ conditions (Fig. 7g). Together, these results suggest that maintaining GLI3 in a repressive state delays satellite cell activation, in part through mTORC1 signaling inhibition. To confirm that GLI3R cooperates with mTORC1 signaling to regulate cell proliferation, FSK and RAPA treatments were also applied on $Gli3^{+/+}$ and $Gli3^{\Delta/\Delta}$ satellite cell-derived myoblasts for 24 h (Fig. 7e). As expected, FSK increases GLI3R levels and inhibits Hedgehog signaling, whereas RAPA inhibits mTORC1 signaling and S6 phosphorylation (Supplementary Fig. 7b–e). Analyzing EdU incorporation revealed that RAPA-mediated inhibition of mTORC1 blocks the proliferation of both $Gli3^{+/+}$ and $Gli3^{\Delta/\Delta}$ myoblasts, while FSK treatment only affects $Gli3^{+/+}$ cell proliferation (Fig. 7h). These findings further implicate GLI3R as acting upstream of mTORC1 signaling to control satellite cell function.

In these experiments, FSK treatment promotes GLI3R activity, consequently inhibiting Hedgehog signaling, diminishing mTORC1 activity, and decreasing the capacity of $Gli3^{+/+}$ cells to activate and proliferate. Accordingly, FSK does not affect $Gli3^{\Delta/\Delta}$ cells. RAPA treatment blocks satellite cell activation and proliferation in all tested conditions. Therefore, our results demonstrate that the Hedgehog mediator GLI3R acts upstream of mTORC1 signaling to control satellite cell quiescence and proliferation.

## Discussion

Quiescent satellite cells lying in a non-cycling, dormant state have a primary cilium. Our study, as well as others[10,41,42], demonstrates that muscle cells are dynamically ciliated as they progress through the myogenic lineage and that the primary cilium reassembles in self-renewing stem cells. Here, we specifically identify the primary cilia-mediated processing of GLI3 as a downstream effector of muscle stem cell regulation. Indeed, satellite cell-specific depletion of GLI3 induces $G_{Alert}$ in QSCs. Moreover, we show that $Gli3$ deletion promotes stem cell expansion and enhances regenerative potential, providing proof of principle for prospective therapeutic applications of our findings.

Our results show that GLI3R colocalizes with PKA at the basal bodies of QSCs. Many studies have delineated the signaling networks that maintain satellite cell quiescence[2,4,5,43–45]. One such pathway, the NOTCH-COLV-Calcitonin receptor (CALCR) cascade, acts through the PKA pathway, which also promotes GLI3 phosphorylation and processing into the repressor form[4,44,45]. Therefore, CALCR-dependent maintenance of $G_0$ in QSCs could be mediated through GLI3 phosphorylation. Upon muscle damage, disruption of the stem cell niche downregulates NOTCH signaling and cAMP-PKA activity[4,44], potentially abrogating GLI3 phosphorylation and processing, which could then promote satellite cell activation independent of Hedgehog ligand-receptor binding. It is highly likely that interaction between signaling networks, such as this, underlies the fine-tuned control of satellite cell activation during regeneration, and further investigations into such interplay will be essential for the development of our understanding of satellite cell biology.

While our study shows that GLI3 is required to maintain a dormant quiescent state, $Gli3$ deletion does not lead to autonomous activation in satellite cells but instead induces an "alert" quiescence. The $G_{Alert}$ state was first described in quiescent stem cells subjected to systemic exposure to HGFA released from a distant muscle injury[6,7]. Interestingly, fully reduced HMGB1 released from a bone injury induces $G_{Alert}$ in SCs through CXCR4 signaling[8], suggesting that several signaling pathways can induce this state in stem cells. In addition, a subpopulation of satellite cells is protected from dioxin through a cell-intrinsic, mTORC1-dependent $G_{Alert}$ response[9], demonstrating that factors released upon injury are not mandatory to induce $G_{Alert}$. Here, we find that the intrinsic loss of GLI3R is sufficient to induce $G_{Alert}$ in satellite cells in the absence of any systemic or extrinsic cues. Processing of GLI3 is likely to be responsive to both extrinsic and intrinsic cues, such as the CALCR discussed above. Hence, we propose that cilia-mediated GLI3 processing controls muscle stem cell quiescence and regulates the first step of entering $G_{Alert}$ in a physiological context.

Intriguingly, our RNA-sequencing data indicates that deleting $Gli3$ from the QSCs and ASCs does not markedly impact the expression of the putative GLI-target genes of the canonical Hedgehog pathway, notably $Gli1$ or $Ptch1$. This is likely due to the heterogeneity of our FACS-sorted cell populations, which could undermine potential variations in the expression of GLI-target genes[30], as QSCs are partially activated[26–28] and ASCs contain

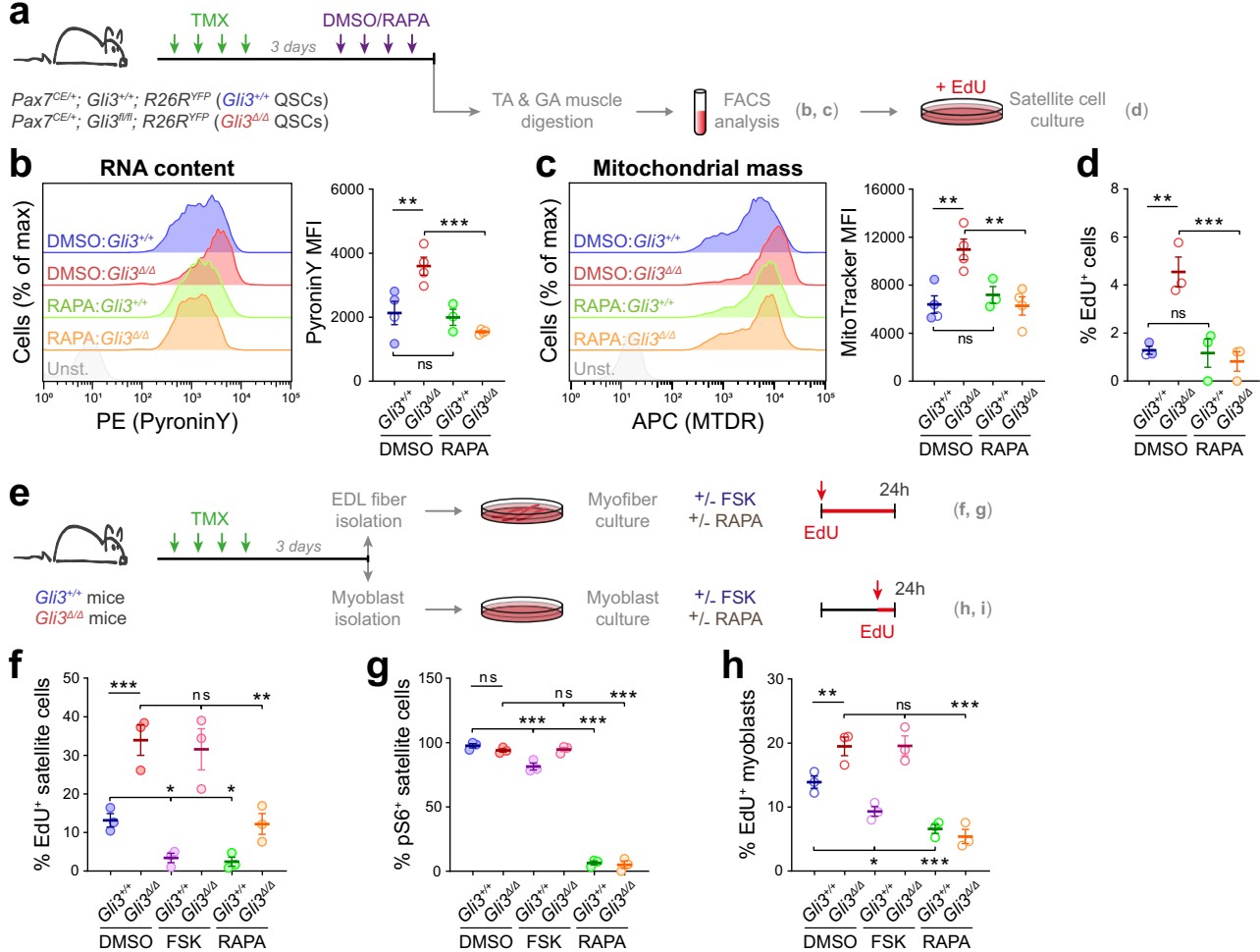

**Fig. 7 GLI3R cooperates with mTORC1 signaling to regulate satellite cell function. a** Experimental design. $Gli3^{+/+}$ and $Gli3^{\Delta/\Delta}$ mice were treated with Rapamycin (RAPA) and QSCs were analyzed for RNA content and mitochondrial mass by FACS and plated 40 h with EdU to analyze cell cycle entry. **b** Left, Representative plots of $Gli3^{+/+}$ and $Gli3^{\Delta/\Delta}$ QSCs isolated from DMSO- or RAPA-treated mice and stained for PyroninY (Unst. unstained); Right, MFI of PyroninY staining. **c** Left, Representative plots of $Gli3^{+/+}$ and $Gli3^{\Delta/\Delta}$ satellite cells isolated from DMSO- or RAPA-treated mice and stained for MitoTracker Deep Red (MTDR). Right, MFI of MTDR staining (**b**, **c**, $n = 4$ mice for DMSO-$Gli3^{+/+}$, DMSO- $Gli3^{\Delta/\Delta}$, RAPA- $Gli3^{\Delta/\Delta}$ and $n = 3$ mice for RAPA-$Gli3^{+/+}$). **d** Proportion of EdU$^+$ satellite cells 40 h post-isolation ($n = 3$ mice). **e** Experimental design. EDL myofibers and myoblasts were isolated from $Gli3^{+/+}$ and $Gli3^{\Delta/\Delta}$ mice and cultured in presence of forskolin (FSK) or RAPA for 24 h. Myofibers were treated with EdU for 24 h post-isolation. Myoblasts received 1h-EdU pulse post-treatment. **f** Proportion of EdU$^+$ satellite cells ($n = 3$ mice). **g** Proportion of pS6$^+$ satellite cells ($n = 3$ mice). **h** Proportions of EdU$^+$ myoblasts in $Gli3^{+/+}$ and $Gli3^{\Delta/\Delta}$ cultures upon FSK and RAPA treatments ($n = 3$ biologically independent samples). Empty dots are males and colored dots are females. Means ± SEM; One-way ANOVA test with Fisher's LSD for multiple comparisons for panels **b**, **c**, **d**, **f**, **g**, **h**, $^*p < 0.05$; $^{**}p < 0.01$; $^{***}p < 0.001$.

both proliferating SCs and myogenic progenitors[29]. Indeed, *Gli1* and *Ptch1* are upregulated in the homogeneous population of $Gli3^{\Delta/\Delta}$ myoblasts. However, canonical Hedgehog signaling cannot be solely responsible for the $Gli3^{\Delta/\Delta}$ phenotype, since the expression of its canonical downstream target genes is mainly turned off when QSCs activate. Instead, mTORC1 activation upon GLI3R depletion is probably responsible for enhanced muscle regeneration[6,38–40].

The upregulation of the GLI transcription factors in ischemia/reperfusion injury protects muscle tissue through activation of AKT/mTOR/p70S6K signaling[46]. Several studies have highlighted the existence of a GLI-mediated mTORC1 activation, where GLI transcription factors regulate positively mTORC1 signaling by downregulation of negative or upregulation of positive mTORC1 mediators[47]. Both $Gli3^{+/+}$ and $Gli3^{\Delta/\Delta}$ QSCs and ASCs display decreased expression of *Deptor*, a negative regulator of mTORC1. Although *Deptor* is not a direct target of GLI3, downregulation of *Deptor* in $Gli3^{\Delta/\Delta}$ SCs is consistent with increased mTORC1

activity and S6 phosphorylation, consequently leading to $G_{Alert}$ transition and enhanced proliferation in $Gli3^{\Delta/\Delta}$ satellite cells[6,38,40]. Thus, we propose the existence of a crosstalk between GLI3-mediated Hedgehog and mTORC1 signaling that controls satellite cell function.

As previously shown[48], the absence of GLI3 in myogenic progenitors increases their proliferation at the expense of early differentiation. However, while *Gli3* deletion in differentiated muscle cells delays the overall muscle regeneration[48], *Gli3* deletion in satellite cells leads to the opposite. $Gli3^{\Delta/\Delta}$ satellite cells form larger myotubes upon differentiation in vitro and promote myofiber hypertrophy in vivo. Likely, this discrepancy is the result of the expansion of satellite cells at the early stages of regeneration/myogenesis, providing a larger pool of progenitor myoblasts that eventually, after a delay, fuse into myofibers or myotubes. The expansion is mediated both by the $G_{Alert}$ state, conferring accelerated proliferative properties to satellite cells, and the increased propensity for self-renewal divisions.

Ultimately, the initial expansion compensates for the delay in myogenic commitment, leading to better repair in $Gli3^{\Delta/\Delta}$ mice.

Different studies have explored the beneficial effects of activating canonical Hedgehog signaling to induce myogenic progenitor proliferation and differentiation during muscle repair[46,49–53]. Increased expression of $Gli1$ and $Ptch1$ was observed in $Gli3^{\Delta/\Delta}$ myoblasts (or myogenic progenitors), consistent with the relief of GLI3R-induced Hedgehog inhibition. On day 5 following CTX-induced acute injury, when the population of myogenic progenitors is predominant, DHH is robustly expressed by the Schwann cells[25]. Thus, the induction of DHH might contribute to the proliferation boost of myogenic progenitors at this stage, by inhibiting GLI3 processing. Our results imply that GLI3R has pleiotropic roles during muscle stem cell progression through the myogenic lineage: controlling satellite cell quiescence and activation independently of Hedgehog ligand-receptor binding, while repressing Hedgehog signaling target genes in myogenic progenitors to regulate their proliferation and differentiation.

Our findings suggest that the primary cilium-mediated control of GLI3 processing regulates muscle stem cell function. However, studies regarding the ablation of primary cilia from muscle cells have led to different and somewhat conflicting results[10,41]. In the C2C12 myogenic cell line, $Ift88$ knockdown results in increased proliferation at the expense of differentiation[41]. Conversely, drug-mediated cilia disassembly does not affect satellite cell proliferation and differentiation but impairs self-renewal on cultured myofibers[10]. This discrepancy is likely related to the experimental settings and cell types. However, these data suggest that the primary cilium has a broader signaling function in muscle cells than the only regulation of GLI3 proteolytic processing.

Although we largely observed beneficial effects of GLI3 depletion, we cannot exclude that long-term loss of GLI3R function and mTORC1-related activation can have negative consequences, as observed in other knockout models[54–56]. $Gli3^{\Delta/\Delta}$ regenerated muscles following triple injury exhibit signs of histological defects, suggesting that permanent GLI3R depletion may eventually lead to regenerative deficits over time. Along these lines, given the importance of quiescence regulation in preventing precocious activation and maintaining the satellite cell pool over an organism's lifetime, one could speculate that the negative consequences of $Gli3^{\Delta/\Delta}$ would not be observed until much later ages. As well, the persistence of a $G_{Alert}$ state in uninjured homeostasis represents an inefficient use of energetic and substrate resources within the muscle, which could be detrimental in contexts of metabolic scarcity[56]. Therefore, translational studies should aim to determine the benefits of transient GLI3 depletion or the pharmacological inhibition of its proteolytic cleavage for the development of stem cell-based therapeutic strategies in regenerative medicine[43,54,57,58].

Our findings represent a seminal advancement in our understanding of the molecular regulation of adult muscle stem cell function. This study establishes that GLI3R controls the transition from $G_0$-to-$G_{Alert}$, as well as the self-renewal division that follows satellite cell activation. As a result, primary cilia-mediated GLI3 processing into a repressor directly impacts the regenerative capacity of stem cells. Further studies are needed to understand how endogenous signaling events or mechanosensing by the primary cilia can control GLI3 processing in response to injury. Finally, pharmacological methods of manipulating GLI3 processing warrant investigation for their potential use in muscle stem cell-based therapies.

## Methods

**Mouse strains and animal care**. Housing, husbandry, and all experimental protocols for mice used in this study were performed in accordance with the guidelines established by the University of Ottawa Animal Care Committee, which is based on the guidelines of the Canadian Council on Animal Care (CCAC). Protocols were approved by Animal Research Ethics Board (AREB) at the University of Ottawa. The following mouse lines were used in this study: $Gli3^{fl/fl}$ mice[59], $Pax7^{CreERT2/+}$ mice[60] referred to as $Pax7^{CE/+}$ in the text, $R26R^{EYFP}$ mice[61] referred to as $R26R^{YFP}$, and $Pax7nGFP$ mice[62]. All the mice used in this study were males and females, from 2 to 10 month-old, with mixed genetic backgrounds (129SV and C57BL/6). Mice were sex and age-matched in all experiments. Animals were group-housed on a 12-h:12-h light:dark cycle and regulated temperature (68–79 °F) and humidity levels (30–70%) and fed ad libitum.

**Tamoxifen injections, Rapamycin treatment, and muscle injury**. Both $Pax7^{CE/+};Gli3^{+/+};R26R^{YFP}$ and $Pax7^{CE/+};Gli3^{fl/fl};R26R^{YFP}$ mice were injected intraperitoneally with 100 µL of a 20 mg mL$^{-1}$ tamoxifen solution (TMX, Sigma T5648) dissolved in corn oil for 4 consecutive days, and then they were maintained on a diet containing tamoxifen (500 mg TMX per kg diet, Teklad, Envigo). Muscle injury was induced by intramuscular injections of cardiotoxin (CTX; 10 µM; Latoxan). Mice were administered buprenorphine and then, anesthetized by isofluorane inhalation. For histological analysis, 50 µL of was injected into the tibialis anterior (TA) muscle. For flow cytometry and FACS analysis, both TA and $gastrocnemius$ muscles were injected with 40 and 80 µL of CTX, respectively. For the Rapamycin treatment, TMX-treated mice were injected intraperitoneally for 4 consecutive days with 100 µL per kg of body weight of either a 20 mM Rapamycin (Tocris) solution or vehicle (10% DMSO diluted in 0.9% NaCl).

**Muscle fixation and histological analysis**. Mice were euthanized and TA muscles were harvested, weighed, and embedded in OCT, and frozen in liquid nitrogen-cooled isopentane. Embedded muscles were transversely sectioned at 10 µm thickness. Sections were post-fixed in 4% PFA/PBS 10 min at room temperature, permeabilized in 0.1 M glycine, 0.1% Triton X-100 in PBS, and blocked in 5% goat serum, 2% BSA in PBS supplemented with M.O.M. Blocking reagent (Vector Laboratories). Then, sections were incubated with primary antibodies as described in "Immunostaining on cells, myofibers, and sections". Muscle cross-sections stained with anti-Laminin and anti-Dystrophin and counterstained with DAPI were analyzed for fiber counting and minimum Feret's diameter using SMASH[63], and centronuclei per myofiber were quantified using MuscleJ[64].

**Flow cytometry and fluorescence-activated cell sorting (FACS)**. Quiescent satellite cells were obtained from uninjured hindlimb muscles, while activated satellite cells were obtained from CTX-injured tibialis anterior and gastrocnemius muscles 3 days after the induced injury. Dissected muscles were minced in collagenase/dispase solution followed by dissociation using the gentleMACS Octo Dissociator with Heaters (Miltenyi Biotec). Satellite cells were sorted by gating a mononuclear cell population of α7-INTEGRIN$^+$, CD34$^+$, CD31$^{neg}$, SCA1$^{neg}$, and CD11b$^{neg}$ (quiescent satellite cells) or α7-INTEGRIN$^+$, VCAM1$^+$, CD31$^{neg}$, CD45$^{neg}$, SCA1$^{neg}$, and CD11b$^{neg}$ (activated satellite cells) using a MoFlo XDP cell sorter (Beckman Coulter). The gating strategy is shown in Supplementary Fig. 10. Flow cytometry analyses (PyroninY, MitoTracker) were performed on a BD LSRFortessa cell analyzer using DIVA software for data collection and FlowJo software for data analysis. The list of antibodies is available in Supplementary Table 1.

*Cell size measurement*. Quiescent and activated satellite cells were sorted based on α7-INTEGRIN$^+$, VCAM1$^+$, CD31$^{neg}$, CD45$^{neg}$, SCA1$^{neg}$, and CD11b$^{neg}$. Single-cell brightfield images were captured on the Amnis ImageStream XMk II and analyzed on the IDEAS Software.

*PyroninY and MitoTracker staining*. About 40 nM PyroninY (Santa Cruz) and 40 nM MitoTracker Deep Red (Thermo Fisher) were added to the muscle digests and incubated for 30 min at 37 °C in water bath. Then, muscle digests were washed and stained with the antibodies for gating the satellite cell population (α7-INTEGRIN$^+$, VCAM1$^+$, CD31$^{neg}$, CD45$^{neg}$, SCA1$^{neg}$, and CD11b$^{neg}$). The gating strategy is shown in Supplementary Fig. 5b.

**In vivo EdU incorporation assay**. Both tamoxifen-treated $Pax7^{CE/+};Gli3^{+/+};R26R^{YFP}$, and $Pax7^{CE/+};Gli3^{fl/fl};R26R^{YFP}$ mice were subjected to CTX-induced muscle injury in both TA and GA muscles. Three days post-injury, mice were injected intraperitoneally with 10 µL per gram of body weight of a 10 mM 5-ethynyl-2′-deoxyuridine (EdU) solution 3 h before sacrifice. Then, muscles were collected and digested as described in "Flow cytometry and fluorescence-activated cell sorting" section.

**EDL myofiber isolation and EdU treatment**. Myofibers were isolated from $extensor$ digitorum longus (EDL) muscles following the previously described protocol[34]. Briefly, EDL muscles were dissected from tendon to tendon and incubated for 1 h in DMEM (Gibco) containing 0.25% collagenase I (Worthington). Single EDL myofibers were isolated by gentle muscle trituration and washed in DMEM. EDL myofibers were finally cultured in DMEM supplemented with 20%

fetal bovine serum, 1% chick embryo extract, and 1% penicillin/streptomycin. Myofibers were fixed at the desired time points using either PFA4%/PBS or ice-cold methanol for immunostaining analysis.

Single EDL myofibers were transfected with *Gli3* siRNA (TriFECTa DsiRNA Kit mouse Gli3, mm.Ri.Gli3.13), *Ift88* siRNA (TriFECTa DsiRNA Kit mouse Ift88, mm.Ri.Ift88.13) or scramble negative control siRNA, at a final concentration of 5 nM using Lipofectamine RNAiMAX (Invitrogen) according to the manufacturer's instructions. siRNA transfection was performed twice at 4 and 16 h post-culture and 6 h after the second transfection, the growth medium was renewed.

For EdU treatment, freshly isolated EDL myofibers were co-treated for 24 h with 20 μM EdU (Thermo Fisher) to analyze satellite cell activation and either 25 μM FSK (Forskolin, R&D Systems) or 100 nM RAPA (Rapamycin, MP Biomedicals). For cell proliferation analysis, 48-h-cultured myofibers were treated with 20 μM EdU for 1 h before fixation.

**Myoblast isolation, culture, and treatments**. Eight- to 16-week-old mice were used to derive primary myoblasts by magnetic cell separation (MACS)[65]. Muscle dissociation and cell filtration was performed following the same protocol described for Flow cytometry and FACS. First, negative lineage selection was performed with biotin-conjugated lineage antibodies (CD11b, SCA1, CD45, and CD31), followed by incubation with streptavidin microbeads. Then, satellite cell-derived myoblasts were purified using a biotin-conjugated anti-α7-INTEGRIN antibody. Myoblasts were cultured on collagen-coated dishes in Ham's F10 medium (Wisent) supplemented with 20% FBS, 1% penicillin/streptomycin, and 5 ng ml$^{-1}$ of basic FGF (Millipore). Differentiation was induced in Ham's F10:DMEM 1:1 supplemented with 5% horse serum, and 1% penicillin/streptomycin.

Primary myoblasts were transfected with *Gli3* siRNA (TriFECTa DsiRNA Kit mouse Gli3, mm.Ri.Gli3.13), *Ift88* siRNA (TriFECTa DsiRNA Kit mouse Ift88, mm.Ri.Ift88.13), or scramble negative control siRNA, at a final concentration of 5 nM using Lipofectamine RNAiMAX (Invitrogen) according to the manufacturer's instructions. siRNA transfection was performed twice, every 24 h, in the growth medium. Cells were collected 48 h after the second transfection.

For GLI3 proteolytic processing analysis, primary myoblasts were treated with either 25 μM FSK (Forskolin, R&D Systems) or 100 nM RAPA (Rapamycin, MP Biomedicals) for 24 h. Equivalent amounts of DMSO were added to the control conditions. For cell proliferation analysis, myoblasts were treated with 20 μM EdU for 1 h before fixation.

**Reserve cell isolation**. Primary myoblasts were derived from 8–16-week-old *Pax7-nGFP* mice and cultured as described above. Myoblasts were differentiated for 3 days. Then, total cells were trypsinized, washed in PBS, and resuspended in FACS buffer. "Reserve" cells were sorted by gating a mononuclear GFP$^+$ cell population using the MoFlo XDP cell sorter (Beckman Coulter).

**RNA extraction and quantitative PCR**. Total RNA was extracted from primary myoblasts using the Nucleospin RNA II kit (Macherey-Nagel) and from satellite cells using the ARCTURUS Picopure RNA extraction kit (Thermo Fisher), according to the manufacturers' instructions. Reverse transcription was performed using SuperScript III Reverse Transcriptase (Invitrogen). Gene expression was assessed with iQ SYBR Green Supermix (Bio-Rad) and analysis was performed using the 2$^{-\Delta\Delta Ct}$ method. RT-qPCR were normalized to the housekeeping genes *Ppia*. A list of primers is available in Supplementary Table 2.

**RNA-sequencing and gene expression analysis**. Twelve samples were used for RNA-sequencing analysis: three samples of quiescent satellite cells and three samples of activated satellite cells for each genotype (*Gli3$^{+/+}$* and *Gli3$^{\Delta/\Delta}$*). For each sample of quiescent satellite cells, all hindlimb muscles from two to three mice were combined. Activated satellite cells were isolated from injured TA and gastrocnemius muscles on day 3 post-CTX injections, and each sample corresponded to one mouse.

Library construction was performed with 20 ng of input total RNA using the NEBNext Ultra II Directional RNA Library Prep Kit for Illumina—polyA mRNA workflow (New England Biolabs). The libraries were sequenced with a NextSeq 500 High Output 75 cycle kit (Illumina). RNA-seq reads were mapped to transcripts from GRCm38_GENCODE.vM19 using salmon v0.13.1[66]. Data was loaded into R using the tximport library and the gene/count matrix was filtered to retain only genes with five or more mapped reads in two or more samples. Differential expression was assessed using DESeq2[67]. PCA was performed using the DESeq2 plotPCA function and rlog-transformed count data. Pearson correlation between means of CPM normalized expression for each replicate group was calculated. Expression differences were calculated using the lfcShrink function, applying the apeglm method (v1.6.0)[68]. Gene ontology (GO) analysis and gene set enrichment analysis (GSEA) were performed using goseq (v1.40.0) and fgsea (v1.14.0) R packages, respectively, on the significantly upregulated and downregulated genes (cut-off of 0.05 and absolute fold changes greater than or equal to 1.2) from *Gli3$^{+/+}$* and *Gli3$^{\Delta/\Delta}$* QSCs and ASCs.

**Western blotting**. Whole-cell proteins were extracted in lysis buffer (150 mM NaCl, 25 mM Tris pH7.5, 1% NP-40, 0.5% sodium deoxycholate, and 0.1% SDS)

supplemented with inhibitors of proteases (Roche) and phosphatases (Sigma). Equal amounts of proteins were resolved on SDS-PAGE 4–12% (Bio-Rad) and transferred onto PVDF membranes. Membranes were blocked using 5% non-fat dry milk in TBS-Tween 0.1% (TBST) for 1 h at room and probed with primary antibodies overnight at 4 °C. The list of antibodies is available in Supplementary Table 1. After four washes in TBST, membranes were incubated for 1 h with HRP-conjugated secondary antibodies at 1:5,000 (Bio-Rad). After four more washes, immunoblots were developed by enhanced chemiluminescence. When required, PVDF membranes were stripped in 62.5 mM Tris HCl, pH6.8, 2% SDS, and 0.8% β-mercaptoethanol.

**Immunostaining on cells, myofibers, and sections**. Following fixation in PFA 2%/PBS, cells and myofibers were washed two times in PBS, permeabilized in 0.1 M Glycine, and 0.1% Triton X-100 in PBS for 10 min. For EdU staining, samples were stained using the Click-iT EdU Alexa Fluor 647 Imaging kit (Thermo Fisher), according to the manufacturer's instructions. Then, cells and myofibers were blocked in 5% horse serum, 2% BSA, and 0.1% Triton X-100 in PBS for at least 1 h, and incubated with primary antibodies overnight at 4 °C. The list of antibodies is available in Supplementary Table 1. Samples were washed three times in PBS, incubated 1 h at room temperature with Alexa Fluor-conjugated secondary antibodies at 1:1,000 (Thermo Fisher), washed three times in PBS, and counterstained with DAPI at 1 μg mL$^{-1}$ in PBS before mounting.

For GLI3/acαTUB/PAX7 co-staining, cells and myofibers were first incubated with anti-GLI3 primary antibody and then, Alexa Fluor donkey anti-goat secondary antibody. Following two washes in PBS-Tween 0.1% and PBS, cells and myofibers were incubated with anti-acαTUB and anti-PAX7 primary antibodies followed by Alexa Fluor goat anti-mouse IgG2b and goat anti-mouse IgG1 secondary antibodies.

Full muscle section pictures were taken on a Zeiss Axio Observer.D1 inverted microscope equipped with an EC Plan-Neofluar 10×/0.3 Ph1 M27 objective and stitched together using Fiji software (http://fiji.sc/Fij). Other immunofluorescence pictures were taken with a Zeiss Axio Observer.D1 inverted microscope equipped with either a Plan-Apochromat 20 × /0.8 M27 objective or a Plan-Apochromat 63 × /1.4.Oil DIC M27 objective. For GLI3 localization, images were taken with a confocal Zeiss LSM 880 AiryScan inverted microscope quipped with a Plan-Apochromat 63 × /1.4.Oil DIC M27 objective. Images were processed and analyzed with Zen and FIJI software.

**In situ force measurement**. Muscle force measurements were performed on an Aurora Scientific 300C-LR-FP dual-mode muscle lever system equipped with a 1 N force transducer and 1 cm lever arm. Electrical stimulation was performed using monopolar needle electrodes attached to an Aurora Scientific 701 C High-Power, Bi-Phase Stimulator. Force transducers were calibrated prior to the study using precision weights. Mice were anesthetized using isoflurane inhalation (2% isoflurane, 1 L/min) until recumbent and non-reflexive to pressure on the paw and positioned on a heated pad to maintain their body temperature at 37 °C throughout the procedure. Mice were positioned supine and hindlimb were shaved. A small incision was made above the hallux and the foot was partially degloved to expose the distal insertion of the tibialis anterior tendon up to the tibialis anterior muscle. The cruciate crural ligament was severed to release the tibialis anterior (TA) tendon from the foot.

A pre-tied loop of waxed 3.5 metric suture was attached to the TA tendon using a series of double thumb knots above, below, and through the loop. The suture was secured to the tendon using minimal amounts of cyanoacrylate glue. The skin of the hindlimb was removed up to mid vastus lateralis to expose the TA and the kneecap. The fascia of the TA was cut using spring scissors. The distal insertion of the TA tendon was severed and the TA was gently lifted to release it from the extensor digitorum longus (EDL) muscle and connective tissue. Muscles were kept from drying using physiological saline. The measured hindlimb was secured between the limb clamp and the stage using a 40 mm long 27 g needle inserted through the epiphysis of the femur immediately proximal to the kneecap and directly into a receiving hole in the stage. Clamping was verified by observing no movement of the kneecap following manipulation of the foot and the needle was secured by a hand screw. The pre-tied loop was attached to the hook on the force transducer lever arm and maintained without tension.

Two monopolar needle electrodes were positioned adjacent to the tibial nerve proximal to the kneecap and distal the kneecap adjacent to the EDL muscle. The transducer was retracted to maintain 20 mN of measured tension for an initial 15-min stretching period with 100 ms trains of 0.3 ms, 5 V supramaximal voltage pulses at 1 Hz stimulation every 100 s. Following stretching, muscles were maintained at 20 mN tension and tetanic contractions were measured every 100 s following 200 ms trains of 0.3 ms, 5 V supramaximal voltage pulses at serial frequencies from 1 to 200 Hz. The maximal force was defined by the difference in maximal force measured during stimulation to that of the tension immediately prior to stimulation.

**Statistics and reproducibility**. No statistical method was used to predetermine sample size. The experimental design incorporated user blinding when possible. All experiments were performed with at least three biological replicates, as indicated in

the figure legends. The results are presented as the mean ± standard error of the mean (SEM). For immunoblots (Fig. 2c, d and Supplementary Figs. 2e, h, 6d, 7b) and immunostaining (Fig. 6e and Supplementary Figs. 1c, 2a, b), experiments have been performed at least three independent times with similar results. Statistical evaluation was performed using two-tailed Student's *t*-test tests to calculate differences between two groups (paired for biologically matched samples, and unpaired for unrelated samples) or univariate ANOVA with post hoc test for multiple comparisons, as appropriate (Graphpad Prism®, Source Data file). The number of independent experimental replications is reported in each corresponding figure legend. Unless otherwise described, data are presented as mean ± SEM and *p* value <0.05 was considered as statistically significant. Throughout the manuscript, level of significance is indicated as follows: *$p < 0.05$, **$p < 0.01$, ***$p < 0.001$. The exact *p* values are provided in the Source Data file.

**Reporting summary**. Further information on research design is available in the Nature Research Reporting Summary linked to this article.

## Data availability

The RNA-sequencing data generated in this study have been deposited in the Gene Expression Omnibus under the accession code GSE144871. The raw images for the immunoblots are provided in Supplementary Fig. 9. Source data are provided with this paper. All other data supporting the findings of this study are available from the corresponding author on reasonable request. Source data are provided with this paper.

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

## Acknowledgements

The authors thank Dr. Valerie Wallace for providing the *Gli3* floxed mice, Jennifer Ritchie for animal husbandry, Fernando Ortiz for FACS, Caroline Vergette from StemCore Laboratories, Gareth Palidwor from Bioinformatics Core, Hani Jrade and Damian Carragher for helping with the ImageStream flow cytometry, Alireza Ghase-mizadeh for helping with confocal imaging, Hong Ming, Ricardo Carmona, and Pascale Muller for technical assistance, and Sandy Martino for administrative assistance. C.E.B. was supported by postdoctoral fellowships from the Ontario Institute for Regenerative Medicine (OIRM) and the French Muscular Dystrophy Association (AFM)-Téléthon and is now supported by a postdoctoral fellowship from the Fondation pour la Recherche Médicale [FRM, ARF201909009155]. A.Y.T.L. is supported by a postdoctoral fellowship from OIRM. P.F. was supported by a doctoral fellowship from the Canadian Institutes of Health Research (CIHR). Studies from the F.L.G. lab are supported by grants from the Agence Nationale pour la Recherche (ANR): Myofuse project [ANR-19-CE13-0016-03] and Myofibrosis project [ANR-19-CE14-0008-02], and from the European Joint Program on Rare Diseases (EJP RD): MYOCITY project. M.A.R. holds a Canada Research Chair in Molecular Genetics. These studies were carried out with the support of grants from the U.S. National Institutes for Health [R01AR044031], the Canadian Institutes of Health Research [FDN-148387], and the Stem Cell Network.

## Author contributions

C.E.B. and M.A.R. conceived the project. C.E.B., M.-C.S., and A.Y.T.L. designed and performed experiments and analyzed the data. D.H. conducted experiments. W.J. performed transcriptomic analysis. P.F. performed force measurement and analysis. C.E.B. wrote the original draft with input from co-authors. C.E.B. and M.A.R. edited the final manuscript. M.A.R. and F.L.G. provided financial support and resources.

## Competing interests

The authors declare no competing interests.
