## [Peer Review File · Nature Communications]

GLI3 Regulates Muscle Stem Cell Entry into GAlert and Self-RenewalREVIEWER COMMENTS

Reviewer #1 (Remarks to the Author):

Summary

The authors identify a critical role for GLI3 and its processing in regulating SC expansion and G0 to GALERT state switching. The study first shows that Gli3 expression is elevated in activated (proliferating) satellite cells, together higher levels of full length GLI3 (GLI3FL). In contrast, the proteolytically processed repressive GLI3 (GLI3R) is more abundant in reserve cells (self-renewed SCs) but low in proliferating myoblasts, coinciding with primary cilia presence and disappearance. These results link GLI3R dependence on the primary cilia. The study then uses siRNA to knock down *Ift88* or Gli3 in satellite cell that are about to divide, both resulting in reduced asymmetric division but increased symmetric expansion of activated satellite cells, suggesting that GLI3FL or GLI3R mainly functions to repress cell expansion (and or maintain quiescence). Using genetics models to study the requirement of GLI3 in Pax7+ cells, the authors find that loss of GLI3 enhances regenerative capacity of muscle after acute and repetitive injuries. Gli3^{-/-} SCs injected into an MDX model also exhibit better engraftment and form more and larger myofibers than did WT SCs, suggesting a potential therapeutic avenue by GLI3 inhibition. The underlying cellular mechanism can be explained by faster cell cycle entry after GLI3 inhibition, as Gli3^{-/-} exhibit typical feature of primed GALERT cells, including increased EdU incorporation, cell size, mitochondrial mass and RNA content. Consistent with previous studies identifying a role for mTORC1 in the generation of SCs into a GALERT state, the authors find that Gli3^{-/-} have an increased % SCs containing phosphorylated S6, supporting the notion that these cells are in a GALERT state. Overall, these data are compelling, highlight a novel therapeutic avenue and expand our knowledge on cilia and cilia-mediated signaling in SCs. However, some concerns should be addressed in a revised version.

Main concerns:

This study primarily uses the genetic mouse model in which Gli3 gene is knockout, which eliminates both the full length GLI3 (GLI3FL) and processed GLI3 (GLI3R), as shown in Figure S3c. Therefore, the model cannot discriminate if the repressor, activator or both functions of GLI3 mediates the observed cellular phenotype. While the *Ift88* siRNA may potentially affect processing of GLI3 and affecting the balance of GLI3FL and GLI3R, such expected results were not clearly demonstrated. Is it possible to quantify the results in Figure S2C to demonstrate alterations in GLI3R/GLI3FL ratios by *Ift88* siRNA? Alternative ways to demonstrate a specific role of GLI3R is to overexpress GLI3R or inhibiting the proteolysis of GLI3FL if feasible. This would also help discern if indeed inhibition of GLI3FL proteolysis constitutes a viable strategy for stem-cell mediated therapies.

Gli3^{-/-} SCs in uninjured muscles appear to be in a GALERT state, suggestive of a role for GLI3R in mediating SC quiescence, however there is no data demonstrating GLI3R protein level in QSCs. Authors should include protein analysis for GLI3 in QSCs (for example by immunofluorescence). Are GLI3R levels more abundant in QSCs? What is the sub-cellular localization of GLI3 in QSCs and ASCs (especially in relative to the cilium)?

The data clearly demonstrate that Gli3^{-/-} SCs act as GALERT cells and enter the cell cycle more rapidly. However, what is the downstream target mediating this effect? Increased pS6 suggests mTOR signaling is activated but whether this is a direct cause of loss of GLI3 or a consequence of GALERT state cannot be discriminated with the current experiments. Did the author look at how Gli3 KO alters expression of conserved GLI target genes?

Minor comments/questions

Based on number of Myog+ cells, the author interpret that Gli3^{-/-} does not affect differentiation. However, given an increase in total Pax7+ cells after Gli3 KO, this may indicate a reduction in differentiation. Please clarify.

Line 213-216 states “Interestingly, the correlation between the QSC and ASC transcriptional signatures was greater in the Gli3 Δ/Δ background than the Gli3^{+/+} (Fig. 6e), suggesting that the Gli3 Δ/Δ QSCs may actually exhibit some features of satellite cell activation.” I noted that the correlations were 0.67 vs 0.66. Are they statistically different?

Figure 2 and Supplementary Figure 2 should have additional explanation in the text about the conclusions drawn from the data. Specifically, Figure 2 is based on SCs cultured on myofibers, while the WB in Supplementary Figure 2 is based on primary myoblasts (which express very low levels of GLI3R). Thus, is the function of GLI3FL primarily being probed with the siRNA experiments? Also the time in Figure 2b is labeled 48h but in the main text it was stated 42 h.

Figures

Figure 1

It would have been nice to include the sub-cellular expression of GLI3 with IF.

Figure 2

OK

Figure 3

-OK

Figure 4

4b and 4c are not consistent as it appears there are more Pax7 cells in the WT in the staining but quantification shows more satellite cells in the KO, perhaps a more representative IF image for 4b should be included.

Figure 5

5a, EDL fiber culture timeline, was the EdU added right before 48 hours or upon isolate? Unclear based off the diagram.

The main text related to this figure states that KO satellite cells remained quiescent (Line 206). If that is the case, how could the satellite cell number increase? Shouldn't they have activated and divided in order to increase in number?

Figure 6

OK

Figure 7

OK

Reviewer #2 (Remarks to the Author):

The manuscript examines the role of the primary cilium and Hedgehog signaling in muscle SCs and how modulation of each affects the properties of the SCs and their ability to regenerate damaged muscle. Most of the experiments are carried out using a Gli3 SC-KO mouse, and the investigators conclude that Gli3 is required to keep SCs in G0, independent of Hedgehog signaling. However, routine studies of Hedgehog signaling are not performed, which would seem to be important controls.

The data shows that Gli3 is processed to a repressive form (Gli3R) in dormant SCs and when the primary cilium is reassembled when myoblasts are in the process of differentiating. This implication is that some signaling through the primary cilium is controlling this processing. This manuscript would be greatly improved by some deeper examination as to what is controlling Gli3 processing, if not Hedgehog signaling, and how the primary cilium is involved in that other signaling pathway.

It is puzzling that a genetic defect in an important signaling pathway produces what the authors describe as “markedly enhanced long-term regeneration” of muscle, as well as “augmented self-renewal” of the SCs themselves. Surely, there must be an overall functional deficit in a cell that has a complete deletion of a key effector in a signaling pathway. It would seem to be important to demonstrate what is deficient in these cells. Many previous reports have demonstrated a role of Hedgehog signaling in myogenic processes (for example: DOI: 10.1016/j.yexcr.2015.10.008; DOI: 10.1038/gt.2014.13; DOI: 10.1074/jbc.M112.400184), so it is difficult to imagine that knocking out Gli3 leads only to an enhancement of SC function.

Since this study is focused on this one SC-KO strain, it is surprising that the studies stop short at the transcriptome. This study would be greatly improved by an analysis of Gli3 target genes that are contributing to the properties of the SCs in these mice. These would be standard validation studies (ChIP-seq analysis, gain or loss of function of downstream targets, etc) rather than just ending with GO terms which have very limited specificity. This would at least contribute an understanding of the how Gli3 is functioning in SCs beyond just descriptions of the SC-KO, and possibly provide data to connect Gli3 to the mTORC1 signaling pathway that activates these cells into the Galert state.

There seems to be conflicting interpretations in terms of the fate of Gli3 KO SCs. The data suggests that the cells favor symmetric division during the proliferative amplification after injury, leading to both an increase in muscle fiber size and an increase in SCs. It would seem that it would have to be one or the other. Is the implication that asymmetric cell division is detrimental for both differentiation and self-renewal? If a mutation favored asymmetric cell division, that would seem to favor an increase in SC number at the expense of cells to generate larger fibers.

Reviewer #3 (Remarks to the Author):

In this manuscript, Brun et. al show that Gli3 is required to maintain quiescence of satellite cells (SC). By using a cleverly designed Gli3 conditional KO mouse model, along with myofiber culture and transplantation experiments, the authors demonstrate that loss of Gli3 increases SCs proliferation resulting in enhanced regeneration and long-term engraftment. Thus, these data suggest that Gli3 might be a potential new target for therapeutic interventions aimed at enhancing muscle repair.

CRITIQUE/COMMENTS/SUGGESTIONS:

1- In the methods section, the authors cite the paper from Corrales et. al “The level of sonic hedgehog signaling regulated the complexity of cerebellar foliation” as the reference to their Gli3 conditional knock-out mouse model. However, this paper created the Gli2 conditional knock-out mouse. The authors should clarify which mouse model was used in this study.

2- The main evidence to “demonstrate that cilia-mediated GLI3 processing regulates asymmetric division”, is presented in Figure S2C. However, the western blot is not convincing to demonstrate that KD of *Ift88* leads to loss of Gli3R. For example, there is uneven loading and there is no Gli3 repressor band visible in the control. Besides repeating this experiment, it might also be useful to probe for Gli1 and *Ptch1* expression (two bona fide Hh transcriptional targets), which should be derepressed in their Gli3 or *Ift88* knock down cells due to loss of Gli3R. Another control would be to treat SCs with SAG, a small molecule Hh agonist, which should activate Hh and block Gli3R repressor formation. In addition, the authors use siRNA mediated KD of Gli3 and *Ift88* to make the connection between Gli3 processing by cilia and to show that Gli3R controls asymmetric vs. symmetric divisions of SCs. However, the authors did not observe any changes in total SCs per fiber (Fig S2d&e), which they see in their genetic model (Fig 5b). Is this due to incomplete knock down or differences in the model?

3- The authors claim these findings are independent of the Hedgehog (HH) pathway. However, they only mention in the discussion that Gli1 and *Ptch1*, two Hh targets frequently being induced upon loss of cilia and loss of Gli3R, are not changed in one of their RNAseq experiments without showing any data. It is possible that just the loss of cilia itself, controlled by the cell cycle state of SCs, could lead to loss of Gli3R. This is a very intriguing model and could even imply that this is a common theme applicable to every ciliated cell entering the cell cycle. To confirm their model, we recommend doing extensive testing of Hh pathway activity via qPCR of Gli1 and *Ptch1* for their critical experiments to confirm their hypothesis that Gli3 is indeed acting “independent of canonical Hedgehog signaling”. It would also rule out the possibility that Hh is being turned off as Gli3 can also act as transcriptional activator under certain circumstances (<https://doi.org/10.1016/j.ydbio.2007.02.029> & [https://doi.org/10.1016/S1534-5807\(03\)00394-0](https://doi.org/10.1016/S1534-5807(03)00394-0)). The authors might already have some RNA samples they could use to run some qPCR. In addition, it would be nice to see some functional experiments to rule this out. The most convincing would, of course, be deleting cilia in SCs in vivo. However, this is beyond the scope of this manuscript. One fairly easy possibility would be to treat Gli3 null SCs with a Hh agonist, which shouldn't have an effect if the hypothesis is true.

4- Some experiments lack adequate numbers to make a definitive conclusion especially when using mice on a mixed background, which received a variable muscle injury:

a. For example, there is a huge spread in the data in Fig 6 D. If the number of animals would have been increased from 3 to 6, would the result hold up?

b. “At 72h, no significant change in the number of MYOG+ cells was observed (Fig. 5e, f). Thus, consistent with our in vivo data, Gli3 deletion increases SC proliferation without affecting their terminal differentiation. “ With higher numbers there actually may be a difference in Myog+ cells and, thus, an effect on terminal differentiation.

c. “Surprisingly, analyzing muscle cross-sections and myofibers showed that Gli3^{+/+} and

Gli3 Δ/Δ SCs are located in their niche, underneath the basal lamina (Supplementary Fig. 5a-c)". This is another example, where the 3 control samples are so widely spread that an increase in sample size could potentially change the conclusion.

5- We commend the authors for using both sexes. However, for some of the mouse experiments in Figures 4 and 5, only 3-4 animals were used per genotype. As there is an intrinsic difference in muscle size based on sex and muscle size is being assessed in those two figures, it would greatly enhance transparency and lend strength to their arguments, if the authors would fully disclose the exact number of males and females used per experiment (maybe even plotting males and females separately?). In addition, we recommend adding the age of the animals to each figure legends as there is quite the spread from young 6 weeks old animals to >5months old.

6- As the isolation procedure of SCs itself can lead to dramatic changes to SCs, it would be nice to see some of the key data points repeated in their beautiful in vivo mouse model. For example, the BrdU or EdU experiment could be done in vivo in combination with the Rosa26-EYFP reporter to look for EYFP+ SCs, which are cycling.

7- The data demonstrating that loss of Gli3 induces proliferation of SCs resulting in more SCs is very clear. However, there is no good explanation for why more SCs lead to better repair after an injury or enhanced engraftment. With the data for Myog+ cells not conclusive (see comment above), it would be nice to look at different time points after injury and look at myogenesis more careful.

a. Similarly, there is no good explanation for the really intriguing transplantation results. Are there more GFP+ myofibers because more GFP+ SCs have fused with existing fibers or are these de novo fibers?

8- Fig 4 and S4: "Even after three rounds of injury, Gli3 Δ/Δ mice maintained a higher number of self-renewing SCs (Fig. 4b, c), increased muscle weight and myofiber hypertrophy compared to controls (Fig. 4d; Supplementary Fig. 4a-c)." Based on Figure S4B, it seems as if there are >33% for myofibers presented. Wouldn't that argue for hyperplasia, which in turn would perfectly explain the increase in TA weight (Fig S4A)?

9- Papers to be discussed in more detail:

a. Fu, et al. 2014 (PNAS) knocked down cilia in cultured myoblasts and found that this increased their proliferation but, at the same time, inhibited myogenesis. Similarly, Jaafer-Marican et al. 2016 (Stem Cell Reports), found that removal of cilia impaired the self-renewal of SCs. Since the authors argue for cilia controlling the repressor state of Gli3, both papers need to be discussed in more detail.

b. In addition, there are several papers showing that ectopically activating Hh signaling enhances adult myogenesis partly by inducing proliferation of SCs [<https://doi.org/10.1111/j.1582-4934.2008.00440.x>, <https://www.nature.com/articles/gt201413>]. The data presented here could therefore also be explained by simple Hh derepression upon loss of Gli3 instead of the CALCR cascade. As mentioned above, it would be nice to show a few more experiments to rule out Hh signaling.

c. Renault et al. 2013 looked at conditional loss of Gli3 in muscle including Gli3 knockout myoblasts (<https://doi.org/10.1161/CIRCRESAHA.113.301546>). It would be nice to

discuss/highlight any overlap or differences in both findings.

10- In the discussion the authors write: "Here, we find that the intrinsic loss of GLI3R is sufficient to induce GAlert in SCs in the absence of any systemic or extrinsic cues, such as Hedgehog signals". However, as mentioned above, there is a clear indication that Hh is being activated upon injury. Without proving that in their experimental setup no Hh ligand is being induced and the pathway is not being activated, this is another very strong statement.

11- The authors show that loss of Gli3 leads to SCs entering a G-ALERT state. It would be nice to see the experiments repeated but this time including the contralateral side, where this phenomenon was first described in Rando's 2014 Nature paper.

12- First sentence of discussion: " QSCs have a primary cilium that maintains them in a non-cycling, dormant state." Since the authors have not deleted cilia in vivo and the presented in vitro knockdown data are insufficient, this is another very strong statement.

13- The authors report in Fig S1a that Gli3 is downregulated in QSCs, which have cilia, and upregulated in ASCs, which don't have cilia. They then show in Fig 2b that the expression levels of Gli1, 2 & 3 are all the same in QSCs, which is different from 2a. This is a little confusing. In addition, the main hypothesis of the authors is that cilia, which are present on QSCs, are required to make Gli3R. However, how does this correlate to the reduction in Gli3 expression (Fig 2a)? Do the authors know that there are sufficient protein levels of Gli3 left to fulfill its function? It might be worth doing some staining for Gli3 and cilia to show that Gli3 is at the ciliary tip in QSCs.

14- It would be nice to see representative images for all the quantifications of Figure 2 C-H. That way the data could be more appreciated.

15- Just FYI, cilia is plural and cilium is singular, thus the title should either be "by primary cilia" or "by the primary cilium".

16- Fig 4f-g: Please be consistent with the nomenclature for YFP vs. GFP (main text vs. figure legend vs. figure).

17- Fig 7C: Please report the number of cells per condition. Also, why are there no error bars?

REVIEWER COMMENTS

We are grateful to the reviewers for appraising our work and for providing detailed feedback and helpful comments for improvement. All of the points have been addressed. We have thoroughly re-organized the figures and the “Results” section in order to include all new findings that support our results. We sincerely hope that these modifications strengthen our manuscript and our conclusion that ciliary GLI3 processing regulates muscle stem cell quiescence and fate.

Reviewer #1 (Remarks to the Author):

Summary

The authors identify a critical role for GLI3 and its processing in regulating SC expansion and G0 to GALERT state switching. The study first shows that Gli3 expression is elevated in activated (proliferating) satellite cells, together higher levels of full length GLI3 (GLI3FL). In contrast, the proteolytically processed repressive GLI3 (GLI3R) is more abundant in reserve cells (self-renewed SCs) but low in proliferating myoblasts, coinciding with primary cilia presence and disappearance. These results link GLI3R dependence on the primary cilia. The study then uses siRNA to knock down *Ift88* or Gli3 in satellite cell that are about to divide, both resulting in reduced asymmetric division but increased symmetric expansion of activated satellite cells, suggesting that GLI3FL or GLI3R mainly functions to repress cell expansion (and or maintain quiescence). Using genetics models to study the requirement of GLI3 in Pax7+ cells, the authors find that loss of GLI3 enhances regenerative capacity of muscle after acute and repetitive injuries. Gli3^{-/-} SCs injected into an MDX model also exhibit better engraftment and form more and larger myofibers than did WT SCs, suggesting a potential therapeutic avenue by GLI3 inhibition. The underlying cellular mechanism can be explained by faster cell cycle entry after GLI3 inhibition, as Gli3^{-/-} exhibit typical feature of primed GALERT cells, including increased EdU incorporation, cell size, mitochondrial mass and RNA content. Consistent with previous studies identifying a role for mTORC1 in the generation of SCs into a GALERT state, the authors find that Gli3^{-/-} have an increased % SCs containing phosphorylated S6, supporting the notion that these cells are in a GALERT state. Overall, these data are compelling, highlight a novel therapeutic avenue and expand our knowledge on cilia and cilia-mediated signaling in SCs. However, some concerns should be addressed in a revised version.

We thank the reviewer for his positive comments and hope that all his concerns are now addressed.

Main concerns

1. This study primarily uses the genetic mouse model in which Gli3 gene is knockout, which eliminates both the full length GLI3 (GLI3FL) and processed GLI3 (GLI3R), as shown in Figure S3c. Therefore, the model cannot discriminate if the repressor, activator or both functions of GLI3 mediates the observed cellular phenotype. While the *Ift88* siRNA may potentially affect processing of GLI3 and affecting the balance of GLI3FL and GLI3R, such expected results were not clearly demonstrated. Is it possible to quantify the results in Figure S2C to demonstrate alterations in GLI3R/GLI3FL ratios by *Ift88* SiRNA? We have now provided quantification of the GLI3FL/GLI3R ratio following *Ift88* siRNA treatment of primary myoblasts and showed that GLI3FL/GLI3R ratio is increased upon *Ift88* siRNA treatment (**Supplementary Fig. 2h-j**). Furthermore, we performed RT-qPCR for *Gli1* and *Ptch1*, two well-known GLI-target genes, and showed that both *Gli3* and *Ift88* siRNA treatments lead to *Gli1* and *Ptch1* upregulation (**Supplementary Fig. 2k, l**). Thus, our results demonstrate that GLI3 acts as a repressor in primary myoblasts and requires the primary cilium to be processed.
2. Alternative ways to demonstrate a specific role of GLI3R is to overexpress GLI3R or inhibiting the proteolysis of GLI3FL if feasible. This would also help discern if indeed inhibition of GLI3FL proteolysis constitutes a viable strategy for stem-cell mediated therapies.

Supporting a specific role for GLI3R in muscle cells, *Gli3* siRNA-treated myoblasts and *Gli3^{Δ/Δ}* primary myoblasts exhibit significant increase in *Gli1* and *Ptch1* expression (**Supplementary Figs. 2l, 6c**), suggesting the GLI3FL is dispensable for target gene-expression and that the repressor function dominates. Thus, these experiments show that GLI3 acts as a repressor in primary myoblasts, even though its expression remains low.

Following the reviewer suggestions, we performed additional experiments using SAG (Smoothed agonist), an agonist of Hedgehog signalling that inhibits GLI3 proteolysis, and FSK (Forskolin), an adenylyl cyclase activator that stimulates PKA activity and promotes subsequent GLI3 phosphorylation and proteolytic cleavage (**Supplementary Figs. 2 and 6d, e**) (Wen *et al.*, 10.1128/MCB.01089-09). In wild-type primary myoblasts, SAG treatment induces a loss of GLI3R, while FSK increases GLI3R levels (**Supplementary Fig. 2a-c**). Consequently, we observed that SAG increases *Gli1* and *Ptch1* expression whereas FSK decreases it. Furthermore, neither SAG nor FSK treatments induce changes in GLI-target gene expression in *Gli3^{Δ/Δ}* primary myoblasts.

Thus, these results support the hypothesis that inhibition of GLI3 proteolytic cleavage is a viable strategy for stem-cell based therapeutic strategies.

3. *Gli3*^{-/-} SCs in uninjured muscles appear to be in a GALERT state, suggestive of a role for GLI3R in mediating SC quiescence, however there is no data demonstrating GLI3R protein level in QSCs. Authors should include protein analysis for GLI3 in QSCs (for example by immunofluorescence). Are GLI3R levels more abundant in QSCs? What is the sub-cellular localization of GLI3 in QSCs and ASCs (especially in relative to the cilium)?

Previous studies have demonstrated that full-length and non-PKA phosphorylated GLI3 (GLI3FL) localizes in the primary cilium, while the cleaved repressor GLI3R is found at the basal body of the cilium where its proteolytic processing occurs (Haycraft *et al.*, 10.1371/journal.pgen.0010053, Tukachinsky *et al.*, 10.1083/jcb.201004108, Wen *et al.*, 10.1128/MCB.01089-09, Fan *et al.*, 10.1016/j.chembiol.2014.10.013). As requested by the reviewer, we analysed GLI3 protein localization by immunofluorescence in satellite cells on freshly isolated and cultured myofibers and on primary myoblasts in proliferation and during differentiation (**Fig. 2**). In ciliated QSCs, GLI3 is not expressed in the primary cilium but localizes at the basal body (**Fig. 2a**). In non-ciliated ASCs, GLI3 localizes to the cytoplasm and around the microtubule-organizing centers (**Fig. 2a**). GLI3 is also found in the cytoplasm in non-ciliated, proliferating primary myoblasts (**Fig. 2b**), but accumulates in the axoneme and the tip of the primary cilium in the few myoblasts that are ciliated. This result is in keeping with our western blot showing that GLI3FL is predominant in these cells (**Fig. 2d-g**). As myoblasts undergo differentiation, GLI3 progressively transits from the axoneme to the basal body of primary cilia and at 72h, most of the PAX7⁺ ‘reserve’ cells have a cilium and maintain high GLI3R localized to the basal body (**Fig. 2b-g**).

In addition to the immunostaining, we have provided quantifications of the number of QSCs, PAX7⁺ myoblasts and reserve cells that display GLI3 staining in the axoneme of the primary cilium (**Fig. 2c**). Throughout this new figure (**Fig. 2**), we provided evidence that GLI3 is localized to the ciliary basal body and processed to the repressor form in QSCs and self-renewing cells, while it is mainly cytoplasmic and expressed as a full-length protein in non-ciliated muscle cells.

4. The data clearly demonstrate that *Gli3*^{-/-} SCs act as GALERT cells and enter the cell cycle more rapidly. However, what is the downstream target mediating this effect? Increased pS6 suggests mTOR signaling is activated but whether this is a direct cause of loss of GLI3 or a consequence of GALERT state cannot be discriminated with the current experiments. Did the author look at how *Gli3* KO alters expression of conserved GLI target genes?

Our RNA-sequencing data did not provide any evidence that the conserved GLI target genes are differentially expressed between *Gli3^{+/+}* and *Gli3^{Δ/Δ}* satellite cells (**Supplementary Fig. 4a-d**) (Kato et al., 10.2174/156652409789105570). To further confirm this result, we performed RT-qPCR on

Gli3^{+/+} and *Gli3*^{Δ/Δ} QSCs and ASCs and found that the two conserved GLI target genes, *Gli1* and *Ptch1*, display similar expression (**Supplementary Fig. 4e, f**). GO term analysis and GSEA did not highlight Hedgehog signalling but instead, revealed other deregulated signalling pathways (**Fig. 4**). Among them, the mTORC1 signalling is activated in both QSCs and ASCs. A few studies have revealed the existence of a GLI-mediated mTORC1 activation (Agarwal *et al.*, 10.1074/jbc.M112.425249, Zeng *et al.*, 10.1159/000484068, Klein *et al.*, 10.1002/ajmg.a.61368, Larsen et Moller, 10.3390/cells9102316), where GLI transcription factors regulate positively mTORC1 signalling by downregulation of negative or upregulation of positive mTORC1 mediators. Both *Gli3*^{+/+} and *Gli3*^{Δ/Δ} QSCs and ASCs display decreased expression of *Deptor*, a negative regulator of mTORC1 (**Supplementary file 2**). Although *Deptor* is not a direct target of GLI3, downregulation of *Deptor* in *Gli3*^{Δ/Δ} satellite cells is consistent with increased mTORC1 activity and S6 phosphorylation.

Determining the GLI3 direct target genes in satellite cells would require ChIP-sequencing. Unfortunately, FACS-based satellite cell isolation does not provide enough material to perform this method. We tried to perform GLI3 ChIP-qPCR on primary myoblasts (see comment #4 reviewer 2), but the experimental outcome is very limited. Indeed, analysing *Gli1* and *Ptch1* expression in *Gli3*^{+/+} and *Gli3*^{Δ/Δ} primary myoblasts revealed that both are upregulated in *Gli3*^{Δ/Δ} primary myoblasts (**Supplementary Fig. 6c**). This suggests that, while there is no variation of canonical Hedgehog target genes in satellite cells, GLI3 binds at least some of its canonical target genes myoblasts. Therefore, ChIP-sequencing in cultured myoblasts would not provide information on GLI3 target genes in satellite cells.

Minor comments/questions

5. Based on number of Myog⁺ cells, the author interpret that *Gli3*^{-/-} does not affect differentiation. However, given an increase in total Pax7⁺ cells after *Gli3* KO, this may indicate a reduction in differentiation. Please clarify.

We agree with the reviewer and we rephrased our interpretation for the *in vivo* results at 7dpi (**Supplementary Fig. 7b-c'**). To get better insights on the cellular mechanisms driven by the loss of GLI3, we increased the number of mice used for the *ex vivo* experiments on cultured myofibers. We confirmed that indeed, *Gli3* deletion delays satellite cell differentiation (**Fig. 7e, f**). As well, we performed *in vitro* differentiation of *Gli3*^{+/+} and *Gli3*^{Δ/Δ} primary myoblasts and showed that deletion of *Gli3* delays early differentiation (**Fig. 7g**). However, this delay is transient and at 72h, *Gli3* deletion eventually leads to larger myotubes due to the initial expansion of the *Gli3*^{Δ/Δ} myoblasts (**Supplementary Fig. 6f**). Together, these data demonstrate that *Gli3* deletion increases satellite cell proliferation at the expense of early differentiation and delays, but does not prevent, terminal differentiation and fusion of myogenic progenitors.

6. Line 213-216 states “Interestingly, the correlation between the QSC and ASC transcriptional signatures was greater in the *Gli3*Δ/Δ background than the *Gli3*^{+/+} (Fig. 6e), suggesting that the *Gli3*Δ/Δ QSCs may actually exhibit some features of satellite cell activation.” I noted that the correlations were 0.67 vs 0.66. Are they statistically different?

They are not statistically different. Although the correlation scores are extremely close, they still indicate a slight difference showing that *Gli3*^{Δ/Δ} QSC signature correlates more with *Gli3*^{Δ/Δ} ASCs compared to the *Gli3*^{+/+} QSC/ASC signatures.

7. Figure 2 and Supplementary Figure 2 should have additional explanation in the text about the conclusions drawn from the data. Specifically, Figure 2 is based on SCs cultured on myofibers, while the WB in Supplementary Figure 2 is based on primary myoblasts (which express very low levels of GLI3R). Thus, is the function of GLI3FL primarily being probed with the siRNA experiments? Also the time in Figure 2b is labelled 48h but in the main text it was stated 42 h.

We first performed the siRNA treatment on primary myoblasts allowing for RNA and protein extraction to test the efficiency of the knockdown by qPCR and Western blot (**Supplementary Fig. 2h-1**). To further characterize the consequences of *Ift88* knockdown, we analysed GLI-target gene expression (see comment #1), indicating that disruption of primary ciliogenesis leads to the same effect as loss of GLI3. GLI3 subcellular localization revealed similar expression patterns in primary myoblasts and activated/proliferating satellite cells (**Fig. 2a, b**). Therefore, we transposed our siRNA protocol from myoblasts to cultured myofibers in order to analyse the consequences of *Ift88* and *Gli3* knockdown on *Myf5^{cre}:Rosa^{YFP}* satellite cells (**Fig. 6**). We have now provided additional explanation in the text about our conclusions and rectified the 42h time in the figure.

Figures

Figure 1

It would have been nice to include the sub-cellular expression of GLI3 with IF.

We have included a subcellular localization analysis of GLI3 in the revised manuscript (**Fig. 2; Supplementary Fig. 2**)

Figure 2

OK

Figure 3

OK

Figure 4

4b and 4c are not consistent as it appears there are more Pax7 cells in the WT in the staining but quantification shows more satellite cells in the KO, perhaps a more representative IF image for 4b should be included.

As recommended, we included a more representative IF picture for the PAX7⁺ cells after triple injury (**Fig. 8k**).

Figure 5

5a, EDL fiber culture timeline, was the EdU added right before 48 hours or upon isolate? Unclear based off the diagram. The main text related to this figure states that KO satellite cells remained quiescent (Line 206). If that is the case, how could the satellite cell number increase? Shouldn't they have activated and divided in order to increase in number?

EdU was added to the medium 1h before fixation (**Supplementary Fig. 6a**). We have clarified this on the figure and in the text. Indeed, we observed a slight increase in the number of satellite cells 2 weeks post-tamoxifen treatment. However, this number remains stable over time, and our results indicate that *Gli3^{Δ/Δ}* satellite cells return to and maintain quiescence under homeostatic conditions (**Fig. 3b**). Satellite cells contribute to uninjured myofiber homeostasis in adulthood (Keefe *et al.*, 10.1038/ncomms8087, Pawlikowski *et al.*, 10.1186/s13395-015-0067-1). Since *Gli3* deletion leads to enhanced cell proliferation and self-renewal, we speculate that the initial depletion of *Gli3* triggers a transient entry into the cell cycle with a preference for self-renewal, increasing the total number of satellite cells before homeostatic quiescence is re-asserted. We have adjusted the text in the “Results” and included a discussion of this hypothesis in the “Discussion” section.

Figure 6

OK

Figure 7

OK

Reviewer #2 (Remarks to the Author):

1. The manuscript examines the role of the primary cilium and Hedgehog signaling in muscle SCs and how modulation of each affects the properties of the SCs and their ability to regenerate damaged muscle. Most of the experiments are carried out using a Gli3 SC-KO mouse, and the investigators conclude that Gli3 is required to keep SCs in G0, independent of Hedgehog signaling. However, routine studies of Hedgehog signaling are not performed, which would seem to be important controls.

We agree with the reviewer that providing a deeper analysis of Hedgehog signalling reinforces our results and helps with discerning the role of Hedgehog signalling in satellite cells *versus* myoblasts/myogenic progenitors. First, we re-investigated our RNA-sequencing on *Gli3*^{+/+} and *Gli3*^{Δ/Δ} quiescent and activated satellite cells. Fold-change analysis, GO term analysis and GSEA did not reveal significant variation of the expression of canonical Hedgehog components or canonical GLI-target genes (**Fig. 4; Supplementary Fig. 4a-d; Supplementary file 2**). To rule out the possibility that canonical Hedgehog signalling is activated in *Gli3*^{Δ/Δ} satellite cells, we then analysed the expression of the two conserved GLI-target genes, *Ptch1* and *Gli1*, in QSCs, ASCs, myoblasts and myotubes (**Supplementary Figs. 4e, f, 6c**). Although *Gli1* and *Ptch1* expression does not vary in *Gli3*^{+/+} and *Gli3*^{Δ/Δ} QSCs and ASCs, *Gli3*^{Δ/Δ} myoblasts display upregulation of *Ptch1* and *Gli1*. This result demonstrates that satellite cells and myoblasts behave differently. While GLI3 does not regulate its canonical target genes in satellite cells, it represses their expression in myoblasts/myogenic progenitors. This is consistent with previous reports showing that activation of canonical Hedgehog signalling participates to the proliferation of myogenic progenitors during adult skeletal muscle repair (Elia *et al.*, 10.1016/j.bbamcr.2007.06.006, Straface *et al.*, 10.1111/j.1582-4934.2008.00440.x, Renault *et al.*, 10.1161/CIRCRESAHA.113.301546, Piccioni *et al.*, 10.1038/gt.2014.13).

Therefore, our results imply that GLI3R has pleiotropic roles during muscle stem cell progression through the myogenic lineage: 1) it controls satellite cell quiescence and activation independently of its canonical target genes, 2) it represses Hedgehog signalling target genes in myogenic progenitors to regulate their proliferation and differentiation.

2. The data shows that Gli3 is processed to a repressive form (Gli3R) in dormant SCs and when the primary cilium is reassembled when myoblasts are in the process of differentiating. This implication is that some signaling through the primary cilium is controlling this processing. This manuscript would be greatly improved by some deeper examination as to what is controlling Gli3 processing, if not Hedgehog signaling, and how the primary cilium is involved in that other signaling pathway.

Several studies have already shown that GLI3 is sequentially phosphorylated at the base of the cilium, first by the cAMP-dependent protein kinase A (PKA) (Hammerschmidt *et al.*, 10.1101/gad.10.6.647, Tiecke *et al.*, 10.1016/j.ydbio.2007.02.017, Niewiadomski *et al.*, 10.1016/j.celrep.2013.12.003, Li *et al.*, 10.1016/j.ydbio.2017.06.035). GLI3 phosphorylation promotes its proteolytic processing into a repressor. Accordingly, decreased PKA activity leads to the activation of Hedgehog signalling that, interestingly, occurs independently of Hedgehog ligand-receptor binding. This information and related publications were added in the “Introduction” section.

To assess whether PKA regulates GLI3 processing in muscle cells, we treated primary myoblasts with forskolin (FSK), an adenylyl cyclase that stimulates PKA activity. By immunostaining and WB, we showed that FSK-mediated PKA-stimulation induces GLI3 processing to a repressor (**Supplementary Fig. 2a-e**). Further confirming this result, *Gli1* and *Ptch1* expression was significantly down-regulated in FSK-treated myoblasts (**Supplementary Fig. 2g**). Of note, FSK treatment on *Gli3*^{Δ/Δ} myoblasts did not affect *Gli1* and *Ptch1* expression, indicating that GLI1 and GLI2 do not compensate for the absence of GLI3 in primary myoblasts (**Supplementary Fig. 6b**).

Finally, we showed that both GLI3R and PKA co-localize at the basal body of the quiescent satellite cells (**Fig. 2a**), supporting the idea that PKA regulates GLI3R processing in muscle stem cells, as

previously shown in other cell types (Tuson *et al.*, 10.1242/dev.070805, Li *et al.*, 10.1016/j.ydbio.2017.06.035).

3. It is puzzling that a genetic defect in an important signaling pathway produces what the authors describe as “markedly enhanced long-term regeneration” of muscle, as well as “augmented self-renewal” of the SCs themselves. Surely, there must be an overall functional deficit in a cell that has a complete deletion of a key effector in a signaling pathway. It would seem to be important to demonstrate what is deficient in these cells. Many previous reports have demonstrated a role of Hedgehog signaling in myogenic processes (for example: DOI: 10.1016/j.yexcr.2015.10.008; DOI: 10.1038/gt.2014.13; DOI: 10.1074/jbc.M112.400184), so it is difficult to imagine that knocking out *Gli3* leads only to an enhancement of SC function.

We thank the reviewer for raising this interesting point. Although we largely observed beneficial effects of GLI3 depletion, our experiments were performed on young and adult mice (< 10 months) and in acute injury settings. Therefore, we cannot exclude the possibility that long-term loss of GLI3 repressor function can have negative consequences. Indeed, although *Gli3*^{Δ/Δ} regenerated muscles are bigger following the triple injury, some areas exhibit clear histological defects with fibrotic areas and tiny myofibers, suggesting that GLI3 depletion may eventually lead to detrimental effects over time following repetitive acute injuries (**Supplementary Fig. 7k**). Along these lines, given the importance of quiescence regulation in preventing precocious activation and maintaining the satellite cell pool over an organism’s lifetime, one could speculate that the negative consequences of *Gli3*^{Δ/Δ} would not be observed until much later ages (> 1 year). Finally, the persistence of a G_{Alert} state in uninjured homeostasis represents an inefficient use of limited energetic and substrate resources within muscle, which could be detrimental in a survival setting. We have included a section in the discussion on these possibilities, but testing the long-term effects of *Gli3* deletion experimentally is, in our opinion, a subject of future studies.

4. Since this study is focused on this one SC-KO strain, it is surprising that the studies stop short at the transcriptome. This study would be greatly improved by an analysis of *Gli3* target genes that are contributing to the properties of the SCs in these mice. These would be standard validation studies (ChIP-seq analysis, gain or loss of function of downstream targets, etc) rather than just ending with GO terms which have very limited specificity. This would at least contribute an understanding of the how *Gli3* is functioning in SCs beyond just descriptions of the SC-KO, and possibly provide data to connect *Gli3* to the mTORC1 signaling pathway that activates these cells into the Galert state.

We agree that ChIP-sequencing would have allowed us to determine the list of GLI3-target genes in satellite cells and to clarify the molecular mechanisms contributing to the G_{Alert} transition in *Gli3*^{Δ/Δ} QSCs. However, there are significant technical limitations that we have encountered in our attempts to perform these studies. ChIP-seq requires an amount of chromatin material not feasibly obtained from FACS-based satellite cell isolation of all posterior hindlimb muscles from one mouse (see comment #4 reviewer 1). Thus, we performed ChIP-qPCR on primary myoblasts as they provide, at low cost, unlimited chromatin material. We tested 4 different antibodies for GLI3 (see Figure below) and analysed GLI3 binding on *Gli1* (positive control), *Myf5* and *Myod1* (two myogenic regulatory factors containing GLI-binding sites and expressed in myoblasts (Gustafsson *et al.*, 10.1101/gad.940702, Voronova *et al.*, 10.1074/jbc.M112.400184). 500μg of chromatin was used per chromatin immunoprecipitation (ChIP) and 5μg of antibodies were added to the chromatin [Protocol from (Addicks *et al.*, 10.1038/s41467-019-12086-9)].

Supporting a role for the repressive form of GLI3 (GLI3R) in primary myoblasts, *Gli3*^{Δ/Δ} myoblasts exhibit increased *Gli1* mRNA level (**Supplementary Fig. 6c**). Therefore, we would have expected that GLI3R binds to the *Gli1* locus to repress its expression in primary myoblasts. However, none of the four antibodies tested indicate a significant enrichment at the *Gli1* locus compared to the IgG controls and the control loci (*IgH* and *globin* genes) (see Figure R1 below). As well, no binding was observed on the *Myf5* and *Myod1* loci previously identified.

Altogether, our results indicate that finding GLI3 specific target genes in satellite cells would require first, ChIP-qPCR optimization in primary myoblasts with an efficient anti-GLI3 antibody and then, the optimization of the ChIP protocol on small amounts of cells.

Figure R1. ChIP-qPCR analysis. Light grey bars represent genomic regions precipitated with IgG-specific antibodies and dark grey bars represent the genomic regions precipitated with GLI3 antibodies (5 μ g for each). *Gli1* is used as a positive control. *IgH* and *Globin* are used as negative controls. (n = 3; Error bars, SEM)

To get a better understanding of GLI3 function in satellite cells, we performed GSEA analysis to screen potential pathways activated in *Gli3* deleted satellite cells. Among the activated pathways, we found the mTORC1 signalling activated in both QSCs and ASCs. A few studies have revealed the existence of a GLI-mediated mTORC1 activation (Agarwal *et al.*, 10.1074/jbc.M112.425249, Klein *et al.*, 10.1002/ajmg.a.61368, Larsen et Moller, 10.3390/cells9102316), where GLI transcription factors regulate positively mTORC1 signalling. Both *Gli3*^{+/+} and *Gli3* ^{Δ/Δ} QSCs and ASCs display decreased expression of *Deptor* (Supplementary Data file 2), a negative regulator of mTORC1 (Laplante et Sabatini, 10.1016/j.cell.2012.03.017). Although *Deptor* has not been described as a direct target of GLI3, downregulation of *Deptor* is consistent with increased mTORC1 activity and G_{Alert} transition in *Gli3* ^{Δ/Δ} QSCs as well as enhanced proliferation in *Gli3* ^{Δ/Δ} ASCs (Rodgers *et al.*, 10.1038/nature13255, Rion *et al.*, 10.1242/dev.172460).

5. There seems to be conflicting interpretations in terms of the fate of Gli3 KO SCs. The data suggests that the cells favor symmetric division during the proliferative amplification after injury, leading to both an increase in muscle fiber size and an increase in SCs. It would seem that it would have to be one or the other. Is the implication that asymmetric cell division is detrimental for both differentiation and self-renewal? If a mutation favored asymmetric cell division, that would seem to favor an increase in SC number at the expense of cells to generate larger fibers.

Our results indicate that *Gli3* depletion favours symmetric expansion at the outset of regeneration. Indeed, we observe expansion of the satellite cell pool immediately following *Gli3* deletion (**Fig. 3a, b**) and following acute injury (**Fig. 8b-e**). As well, the *ex vivo* fibre studies demonstrate that *Gli3^{Δ/Δ}* satellite cells favour symmetric division during the first division following activation (**Fig. 6e, i**). In keeping with this, we observed a delay in differentiation of *Gli3^{Δ/Δ}* satellite cells that is consistent with a propensity for self-renewal (**Fig. 8**). However, it is also clear that *Gli3^{Δ/Δ}* satellite cells are not completely deficient of differentiation potential and do eventually differentiate (**Fig. 7e-g, Supplementary Fig. 6f**). Therefore, we believe that, although there is a delay in commitment, the initial expansion of satellite cells provides a larger pool of progenitors that eventually terminally differentiate, thus leading to the observed increase in both fibre size and self-renewal. We have clarified our interpretation in the text of the “Results” and “Discussion” sections.

Reviewer #3 (Remarks to the Author):

In this manuscript, Brun et. al show that Gli3 is required to maintain quiescence of satellite cells (SC). By using a cleverly designed Gli3 conditional KO mouse model, along with myofiber culture and transplantation experiments, the authors demonstrate that loss of Gli3 increases SCs proliferation resulting in enhanced regeneration and long-term engraftment. Thus, these data suggest that Gli3 might be a potential new target for therapeutic interventions aimed at enhancing muscle repair.

We thank the reviewer for the appreciation of our work.

CRITIQUE/COMMENTS/SUGGESTIONS:

1- In the methods section, the authors cite the paper from Corrales et. al “The level of sonic hedgehog signaling regulated the complexity of cerebellar foliation” as the reference to their Gli3 conditional knock-out mouse model. However, this paper created the Gli2 conditional knock-out mouse. The authors should clarify which mouse model was used in this study.

We mistakenly cited the paper from Corrales *et al.* and we have now cited the right publication from Blaess *et al.*, Development, 2008 corresponding to the *Gli3* flox allele used in this study.

2- The main evidence to “demonstrate that cilia-mediated GLI3 processing regulates asymmetric division”, is presented in Figure S2C. However, the western blot is not convincing to demonstrate that KD of *Ift88* leads to loss of Gli3R. For example, there is uneven loading and there is no Gli3 repressor band visible in the control. Besides repeating this experiment, it might also be useful to probe for *Gli1* and *Ptch1* expression (two bona fide Hh transcriptional targets), which should be derepressed in their Gli3 or *Ift88* knock down cells due to loss of Gli3R. Another control would be to treat SCs with SAG, a small molecule Hh agonist, which should activate Hh and block Gli3R repressor formation. In addition, the authors use siRNA mediated KD of Gli3 and *Ift88* to make the connection between Gli3 processing by cilia and to show that Gli3R controls asymmetric vs. symmetric divisions of SCs. However, the authors did not observe any changes in total SCs per fiber (Fig S2d&e), which they see in their genetic model (Fig 5b). Is this due to incomplete knock down or differences in the model?

We repeated the Western blot and as suggested by the reviewer #2, we have quantified the levels and ratio of GLI3FL/GLI3R (**Supplementary Fig. 2h-j**). As well, we performed RT-qPCR for the two *bona fide* Hh transcriptional targets, namely *Gli1* and *Ptch1* (**Supplementary Fig. 2k, l**). We show that both *Gli3* and *Ift88* knockdown results in increased expression of *Gli1* and *Ptch1*, confirming that GLI3 acts as a repressor in muscle cells and that the primary cilium is essential for this processing.

As suggested, myoblasts were also treated with SAG (Smoothed agonist) and forskolin (FSK, an adenylyl cyclase activator that stimulates PKA) (**Supplementary Fig. 2a-g**). We analysed by Western blotting and immunostaining GLI3 proteolytic cleavage and localization in the treated cells (**Supplementary Fig. 2a-c**). SAG treatment induces the accumulation of GLI3FL at the tip of the cilium and abrogates its proteolytic cleavage. However, FSK treatment blocks GLI3 accumulation to the cilium and promotes GLI3 processing to a repressor. In addition, we showed that *Gli1* and *Ptch1* were upregulated in SAG-treated myoblasts and downregulated in FSK-treated myoblasts (**Supplementary Fig. 2f, g**). Altogether, this new dataset indicates that GLI3R regulates canonical GLI-target genes in primary myoblasts.

We also performed RT-qPCR on *Gli3^{Δ/Δ}* myoblasts, showing that *Gli1* and *Ptch1* are upregulated in these cells compared to the *Gli3^{+/+}* myoblasts (**Supplementary Fig. 6c**). Strikingly, SAG and FSK treatments failed to change the two GLI-target gene expression, *Ptch1* and *Gli1*, indicating that GLI1 and GLI2 cannot compensate for the absence of GLI3 (**Supplementary Fig. 6d, e**).

Regarding the experiments done with *Myf5^{Cre}:Rosa^{YFP}* myofibers, we did not observe an increase in the total number of satellite cells per fibre following *Gli3* and *Ift88* knockdown (**Fig. 6f, j**), while we observe a higher number of satellite cells on *Gli3^{Δ/Δ}* myofibers (**Fig. 7b**). This difference is first attributed to the time point analysed. *Myf5^{Cre}:Rosa^{YFP}* myofibers are fixed at 42h, right after the first satellite cell division occurs. Following the first division, satellite cells begin to proliferate rapidly, which is when the *Gli3^{Δ/Δ}* myofibers were fixed. Second, the increase of *Gli3^{Δ/Δ}* satellite cells per myofiber results from a cumulative effect of the G_{Alert} , allowing faster entry into the cell cycle, and the enhanced ability to proliferate, leading to an increase in the number of divisions that have occurred by the 48h time-point. In contrast, given that *Gli3* and *Ift88* knockdown occurs at the beginning of culturing in growth media and that the fibres are fixed by 42h, it is unlikely there would be sufficient time for multiple cell divisions to occur, as we observed. Finally, as noted by the reviewer, *Gli3* knockdown efficiency is not comparable to a complete knockout. Ultimately, these two experiments are independent and do not seek to answer the same questions. The *Gli3^{Δ/Δ}* myofiber experiment was designed to quantify satellite cell proliferation, while *Myf5^{Cre}:Rosa^{YFP}* knockdown experiment was designed to analyse satellite cell fate decisions during the first division.

3- The authors claim these findings are independent of the Hedgehog (HH) pathway. However, they only mention in the discussion that *Gli1* and *Ptch1*, two Hh targets frequently being induced upon loss of cilia and loss of Gli3R, are not changed in one of their RNAseq experiments without showing any data. It is possible that just the loss of cilia itself, controlled by the cell cycle state of SCs, could lead to loss of Gli3R. This is a very intriguing model and could even imply that this is a common theme applicable to every ciliated cell entering the cell cycle. To confirm their model, we recommend doing extensive testing of Hh pathway activity via qPCR of *Gli1* and *Ptch1* for their critical experiments to confirm their hypothesis that *Gli3* is indeed acting “independent of canonical Hedgehog signaling”. It would also rule out the possibility that Hh is being turned off as *Gli3* can also act as transcriptional activator under certain circumstances (<https://doi.org/10.1016/j.ydbio.2007.02.029> & [https://doi.org/10.1016/S1534-5807\(03\)00394-0](https://doi.org/10.1016/S1534-5807(03)00394-0)). The authors might already have some RNA samples they could use to run some qPCR. In addition, it would be nice to see some functional experiments to rule this out. The most convincing would, of course, be deleting cilia in SCs in vivo. However, this is beyond the scope of this manuscript. One fairly easy possibility would be to treat *Gli3* null SCs with a Hh agonist, which shouldn't have an effect if the hypothesis is true.

To clarify the absence of Hedgehog signalling activation in satellite cells, we have now provided heatmaps showing the expression of canonical Hedgehog components and GLI-target genes and their fold-change in *Gli3^{+/+}* and *Gli3^{Δ/Δ}* QSCs and ASCs (**Supplementary Fig. 4a-d**). We also performed RT-qPCR on *Gli3^{+/+}* and *Gli3^{Δ/Δ}* QSCs and ASCs, confirming that *Gli1* and *Ptch1* expression do not vary (**Supplementary Fig. 4h, f**). As well, we conducted additional analysis of our RNA-sequencing data. Specifically, GO term analysis and GSEA did not reveal any activation of canonical Hedgehog signalling (**Fig. 4**). Altogether, these data demonstrate that loss of GLI3R in satellite cells does not affect the expression of its canonical target genes.

However, when analysing *Gli1* and *Ptch1* expression in primary cells, we found that both were upregulated in *Gli3^{Δ/Δ}* myoblasts (**Supplementary Fig. 6c**), indicating that GLI3R controls the expression of its canonical target genes in these cells. It also suggests that GLI3R behave differently in myogenic progenitors or myoblasts compared to quiescent and activated satellite cells, where it regulates other sets of genes. This was added in the “Discussion” section. Ultimately, performing GLI3 ChIP-sequencing in satellite cells and myogenic progenitors would confirm this hypothesis but will require both a protocol optimization for small amounts of cells and efficient antibody (See comment #4 reviewer 1 and comment #4 reviewer 2). Further delineation of the system is, in our opinion, the subject of future studies.

As suggested, we tested SAG treatment in *Gli3^{Δ/Δ}* primary cells and did not observe effects on *Gli1* and *Ptch1* expression (**Supplementary Fig. 6d**, see comment #2 above).

4- Some experiments lack adequate numbers to make a definitive conclusion especially when using mice on a mixed background, which received a variable muscle injury:
 a. For example, there is a huge spread in the data in Fig 6 D. If the number of animals would have been increased from 3 to 6, would the result hold up?

We have now plotted 6 mice per genotype (3 males and 3 females) and observed a similar effect and variance (**Fig. 3d**).

b. “At 72h, no significant change in the number of MYOG⁺ cells was observed (Fig. 5e, f). Thus, consistent with our *in vivo* data, Gli3 deletion increases SC proliferation without affecting their terminal differentiation. “ With higher numbers there actually may be a difference in Myog⁺ cells and, thus, an effect on terminal differentiation.

As requested by the reviewer, we added 3 male mice per genotype for the *ex vivo* experiments involving Gli3^{+/+} and Gli3^{Δ/Δ} myofibers (**Fig. 7a, b, e, f**). We now show a significant decrease in the proportion of MYOG⁺ cells, indicating an impairment of myogenic differentiation. To further assess their terminal differentiation, primary cells were differentiated *in vitro* (**Fig. 7g; Supplementary Fig. 6f**). Although Gli3^{Δ/Δ} myoblasts exhibit decreased expression of the myogenic regulatory factors, MYOD1, MYOGENIN and Myosin Heavy Chain (MyHC), at the early steps of differentiation (0-24h), they express similar levels of MyHC at the later steps (48-72h). Together, these results showed that Gli3 deletion increases satellite cell proliferation at the expense of early differentiation and delays but does not prevent terminal differentiation and fusion of myogenic progenitors.

c. “Surprisingly, analyzing muscle cross-sections and myofibers showed that Gli3^{+/+} and Gli3^{Δ/Δ} SCs are located in their niche, underneath the basal lamina (Supplementary Fig. 5a-c)”. This is another example, where the 3 control samples are so widely spread that an increase in sample size could potentially change the conclusion.

As suggested by the reviewer, we increased the sample size for both Gli3^{+/+} and Gli3^{Δ/Δ} mice (n = 4 males and 4 females for each genotype) (**Supplementary Fig. 3e, f**). The proportion of satellite cells outside the basal lamina remains unchanged.

5- We commend the authors for using both sexes. However, for some of the mouse experiments in Figures 4 and 5, only 3-4 animals were used per genotype. As there is an intrinsic difference in muscle size based on sex and muscle size is being assessed in those two figures, it would greatly enhance transparency and lend strength to their arguments, if the authors would fully disclose the exact number of males and females used per experiment (maybe even plotting males and females separately?). In addition, we recommend adding the age of the animals to each figure legends as there is quite the spread from young 6 weeks old animals to >5months old.

We have updated the text to address the reviewer’s suggestion. All our mouse experiments were performed on sex- and age-matched mice, which is now indicated in the “Methods” section. We have also indicated the exact number of males and females used per experiment in the figure legends (**Fig. 3b, 3d, 5, 7, 8; Supplementary Figs. 3d-g, 5, 6b, 7**). When absolute numbers are presented, we have also plotted males and females separately. The experiments (tamoxifen IPs, cardiotoxin injuries, etc.) were mainly performed on 6-to-12 week old mice. However, muscles were collected between 3 to 6 months, depending on the experimental design. To enhance transparency, we have provided all the dates of birth and sacrifice of the mice used in this study in the **Data Source file**.

6- As the isolation procedure of SCs itself can lead to dramatic changes to SCs, it would be nice to see some of the key data points repeated in their beautiful *in vivo* mouse model. For example, the BrdU or EdU experiment could be done *in vivo* in combination with the Rosa26-EYFP reporter to look for EYFP⁺ SCs, which are cycling.

We have added an analysis of *in vivo* EdU incorporation at day 3 post-cardiotoxin-induced injury (Fig. 7c, d) and confirmed the increased ability of $Gli3^{\Delta/\Delta}$ ASCs to proliferate. As well, we analysed *in vivo* EdU incorporation in non-injured mice in order to determine whether $Gli3^{\Delta/\Delta}$ satellite cells are cycling in this particular context (see Figure R2 below). Following the protocol of Zismanov *et al.* (10.1016/j.stem.2015.09.020), we did not observe any increase in EdU incorporation in $Gli3^{\Delta/\Delta}$ satellite cells, further confirming that these cells are not activated and remain quiescent.

Figure R2. 7 days after the first tamoxifen injection, $Gli3^{+/+}$ and $Gli3^{\Delta/\Delta}$ mice received 100 μ l of 20mM EdU solution (in PBS) by intraperitoneal injections five times at 8hr intervals ($n = 4 \times 3$ -month old males). EdU⁺ satellite cells (ITGA7⁺, VCAM⁺, Lin⁻) were analysed by flow cytometry.

In these specific experiments, we did not use the Rosa26-EYFP reporter as the cells were fixed, leading to the loss of our YFP signal. The Rosa26-EYFP reporter has been mainly used for the engraftment assay, in order to track both satellite cells and myofibers post-transplantation (Supplementary Fig. 5g-j).

7- The data demonstrating that loss of Gli3 induces proliferation of SCs resulting in more SCs is very clear. However, there is no good explanation for why more SCs lead to better repair after an injury or enhanced engraftment. With the data for Myog⁺ cells not conclusive (see comment above), it would be nice to look at different time points after injury and look at myogenesis more careful.

As suggested, we have performed a deeper analysis of myogenesis in $Gli3^{\Delta/\Delta}$ myogenic progenitors. $Gli3^{\Delta/\Delta}$ primary myoblasts display decreased levels of the myogenic regulatory factors, MYOD1 and MYOGENIN (Fig. 7g, Supplementary Fig. 6c). At the early differentiation time point, $Gli3^{\Delta/\Delta}$ myoblasts display delayed differentiation. However, they retain myogenic differentiation capacity and at 72h differentiation, they form larger myotubes, consistent with our *in vivo* myofiber diameter findings (Fig. 7g, Supplementary Fig. 6f). Our results indicate that $Gli3$ deletion delays, rather than impairs, early myogenic differentiation and does not prevent terminal differentiation. This is consistent with our *in vivo* data showing that MYOG⁺ cells are less abundant at 7dpi, although the overall efficiency of muscle regeneration is not affected. Therefore, we speculate that the expansion of satellite cells at the early stages of regeneration/myogenesis provides a larger pool of progenitor myoblasts that eventually, after a delay, fuse into myofibers or myotubes. We have included an explanation of these results in the “Discussion” section.

a. Similarly, there is no good explanation for the really intriguing transplantation results. Are there more GFP⁺ myofibers because more GFP⁺ SCs have fused with existing fibers or are these de novo fibers?

RNA-sequencing data and phenotypic analysis of QSCs and ASCs demonstrate that $Gli3^{\Delta/\Delta}$ satellite cells maintain stemness and are poised to activate faster, adhere less and proliferate more (Figs. 6-8; Supplementary Figs. 6-8). $Gli3^{\Delta/\Delta}$ QSCs engraft as well as $Gli3^{+/+}$ QSCs, but they proliferate and migrate more in the MDX regenerating environment, consequently increasing, as discussed above, the number of satellite cells and subsequently the number of myogenic progenitors, leading to more cells able to commit to terminal differentiation (Fig. 5f-h; Supplementary Fig. 5h). Thus, the increase in GFP⁺ myofibers likely arises from this initial expansion of the satellite cell pool following engraftment.

8- Fig 4 and S4: “Even after three rounds of injury, $Gli3^{\Delta/\Delta}$ mice maintained a higher number of self-renewing SCs (Fig. 4b, c), increased muscle weight and myofiber hypertrophy compared to controls

(Fig. 4d; Supplementary Fig. 4a-c).” Based on Figure S4B, it seems as if there are >33% for myofibers presented. Wouldn’t that argue for hyperplasia, which in turn would perfectly explain the increase in TA weight (Fig S4A)?

We agree with the reviewer and rephrased our conclusions. We provided representative pictures of the full sections of *Gli3*^{+/+} and *Gli3*^{Δ/Δ} TA injured muscles showing that 1) *Gli3*^{Δ/Δ} TA muscles are bigger, 2) myofibers are overall larger and 3) there are some areas with great amounts of small myofibers and fibrosis (**Supplementary Fig. 7k**). Therefore, *Gli3*^{Δ/Δ} triple-injured mice exhibited increased muscle weight resulting from both myofiber hyperplasia and hypertrophy.

9- Papers to be discussed in more detail:

a. Fu, et al. 2014 (PNAS) knocked down cilia in cultured myoblasts and found that this increased their proliferation but, at the same time, inhibited myogenesis. Similarly, Jaafer-Marican et al. 2016 (Stem Cell Reports), found that removal of cilia impaired the self-renewal of SCs. Since the authors argue for cilia controlling the repressor state of Gli3, both papers need to be discussed in more detail.

We thank the reviewer for his suggestion and have included a discussion of both papers in the “Discussion” section.

b. In addition, there are several papers showing that ectopically activating Hh signaling enhances adult myogenesis partly by inducing proliferation of SCs [<https://doi.org/10.1111/j.1582-4934.2008.00440.x>, <https://www.nature.com/articles/gt201413>]. The data presented here could therefore also be explained by simple Hh derepression upon loss of Gli3 instead of the CALCR cascade. As mentioned above, it would be nice to show a few more experiments to rule out Hh signaling.

We performed RT-qPCR for *Gli1* and *Ptch1*, two canonical GLI-target genes in QSCs, ASCs, myoblasts and 72h-differentiated myoblasts (see comment #3). The CALCR cascade involves PKA activity to regulate muscle quiescence, which is required to process GLI3 as a repressor. Therefore, it seemed important to discuss this potential crosstalk between signalling pathways that should be addressed in future studies.

Our experiments demonstrate that, while canonical Hedgehog signalling is not involved in the *Gli3*^{Δ/Δ} phenotype of QSCs and ASCs, it is activated in primary myoblasts. This is consistent with the studies of Pola’s lab where they analysed Hedgehog signalling in myogenic progenitors and showed that Shh treatment induces increased proliferation of myogenic progenitors at 4 days post-injury (Piccioni *et al.*, 10.1038/gt.2014.13), while cyclopamine treatment decreases their ability to proliferate (Straface *et al.*, 10.1111/j.1582-4934.2008.00440.x). Upon *Gli3* deletion, *Gli1* and *Ptch1* are upregulated in *Gli3*^{Δ/Δ} myoblasts, suggesting activation of canonical Hedgehog signalling. At 3 days post-injury where the muscle cell population represents a mixed population of satellite cells and myogenic progenitors (Oprescu *et al.*, 10.1016/j.isci.2020.100993), *Gli3*^{Δ/Δ} ASCs proliferate more. Although we did not analyse the same injury time points, it is more likely that the Hedgehog signalling activation is more pronounced at day 4 post-injury, when the myogenic progenitor population increases.

c. Renault et al. 2013 looked at conditional loss of Gli3 in muscle including Gli3 knockout myoblasts (<https://doi.org/10.1161/CIRCRESAHA.113.301546>). It would be nice to discuss/highlight any overlap or differences in both findings.

Renault et al. generated *HSA*^{CreERT2};*Gli3*^{fl/fl} mice allowing for *Gli3*-specific deletion in myocytes and myofibers. Consistent with our results *in vitro*, they observed increased proliferation at the expense of myogenic differentiation in their myoblast culture. Consequently, muscle repair induced by hindlimb ischemia (HLI) is delayed following *Gli3* deletion at 10 days post-injury. To explain such differences in our results, we can mention:

a. In their study, *Gli3* was knocked-out in myocytes and myofibers while we knocked-out *Gli3* at earlier lineage stages in quiescent and activated satellite cells as well as myoblasts/myocytes.

- b. We analysed muscle regeneration until complete repair (21 days post-injury) while they analysed 5 and 10 days post-injury only.
- c. We did not perform the same type of injury (hindlimb ischemia *versus* cardiotoxin), which can have major impact on the regeneration outcome (Hardy *et al.*, 10.1371/journal.pone.0147198).

In our mouse model, there is cumulative effects of GLI3 loss in the quiescent and activated satellite cells as well as myoblasts/myocytes. Quiescent satellite cells transit to G_{Alert} and activated satellite cells exhibit enhanced ability to proliferate, two mechanisms promoting accelerated muscle regeneration and that are not visible in $HSA^{CreERT2};Gli3^{fl/fl}$ mice. Therefore, G_{Alert} , increased proliferation and delayed early differentiation promote overall muscle regeneration. We added this paper in the “Discussion” section of our manuscript.

10- In the discussion the authors write: “Here, we find that the intrinsic loss of GLI3R is sufficient to induce G_{Alert} in SCs in the absence of any systemic or extrinsic cues, such as Hedgehog signals”. However, as mentioned above, there is a clear indication that Hh is being activated upon injury. Without proving that in their experimental setup no Hh ligand is being induced and the pathway is not being activated, this is another very strong statement.

We rephrased our sentence as follows: “Here, we find that the intrinsic loss of GLI3R is sufficient to induce G_{Alert} in quiescent satellite cells, in the absence of any systemic or extrinsic cues, such as Hedgehog signals”.

However, we would like to mention that we did not observe any upregulation of *Shh*, *Dhh* or *Ihh* in total resting muscle extract from $Gli3^{\Delta/\Delta}$ mice compared to $Gli3^{+/+}$ mice, indicating that there is no Hh ligand being induced upon *Gli3* deletion in satellite cells (see Figure R2 below).

Figure R3. Expression levels of *Shh*, *Ihh* and *Dhh*, the three Hh ligands, and *Gli1* and *Ptch1*, two Hh target genes, normalized to *Tbp* and *Ppia*, in uninjured TA muscle (n = 3).

Interestingly, DHH is expressed by the Schwann cells at 5 days post-CTX-induced injury, when the number of myogenic progenitors is the highest (Kopinke *et al.*, 10.1016/j.cell.2017.06.035). Thus, DHH may participate to Hedgehog signalling activation in muscle progenitors *in vivo*. We discussed this point in the “Discussion” section.

11- The authors show that loss of *Gli3* leads to SCs entering a G_{Alert} state. It would be nice to see the experiments repeated but this time including the contralateral side, where this phenomenon was first described in Rando’s 2014 Nature paper.

Although this is an interesting suggestion, the G_{Alert} state induced by the HGFA/c-MET/mTORC1 signaling cascade in the leg of the contralateral side has been documented already (Rodgers *et al.*, 10.1038/nature13255, Rodgers *et al.*, 10.1016/j.celrep.2017.03.066). Since we observed the G_{Alert} phenotype in resting, uninjured muscle, we conclude that *Gli3* knockout satellite cells are perpetually in a G_{Alert} state. It is therefore unlikely, in our opinion, that any additional or perceptible phenotype will be observed in the contralateral leg upon injury in *Gli3* knockout mice. Ultimately, while it is possible that

a small additive effect could be observed, such an observation would not significantly expand the conclusions of our study. However, we have included a relevant discussion of Rando's 2014 Nature paper in the "Discussion". Moreover, several studies have highlighted other pathways able to induce G_{Alert} in QSCs without requiring muscle injury (Lee *et al.*, 10.1073/pnas.1802893115, Der Vartanian *et al.*, 10.1016/j.stem.2019.03.019). Therefore, we do not feel that these experiments are warranted at this time.

12- First sentence of discussion: "QSCs have a primary cilium that maintains them in a non-cycling, dormant state." Since the authors have not deleted cilia in vivo and the presented in vitro knockdown data are insufficient, this is another very strong statement.

We agree with the reviewer and we rephrased the sentence as follows: "Quiescent satellite cells lying in a non-cycling, dormant state have a primary cilium."

13- The authors report in Fig S1a that *Gli3* is downregulated in QSCs, which have cilia, and upregulated in ASCs, which don't have cilia. They then show in Fig 2b that the expression levels of *Gli1*, 2 & 3 are all the same in QSCs, which is different from 2a. This is a little confusing. In addition, the main hypothesis of the authors is that cilia, which are present on QSCs, are required to make *Gli3R*. However, how does this correlate to the reduction in *Gli3* expression (Fig 2a)? Do the authors know that there are sufficient protein levels of *Gli3* left to fulfill its function? It might be worth doing some staining for *Gli3* and cilia to show that *Gli3* is at the ciliary tip in QSCs.

Rather than fold change relative to QSCs, we have plotted *Gli1*, *Gli2* and *Gli3* expression relative to reference genes (**Supplementary Fig. 1c**). We updated our heatmap that shows a larger number of components of the Hedgehog signalling and *GLI*-target genes. As well, we provided a second heatmap showing their expression fold change, where significantly deregulated genes are in bold with a star. Together, these data help for visualising Hedgehog-related gene expression compared to the table (**Supplementary Data file 1**) and demonstrate that canonical Hedgehog signalling is turned off during satellite cell activation.

In addition to the transcriptional data, we provided immunostaining for *GLI3* showing its subcellular localization in satellite cells and myoblasts (**Fig. 2a, b**). In particular, *GLI3* is expressed at the basal body of QSCs where it is known to be processed as a repressor. *Gli3* upregulation during myogenic progression may be required to compensate the low level of *GLI3R* in proliferating cells, required to control cell cycle and differentiation.

14- It would be nice to see representative images for all the quantifications of Figure 2 C-H. That way the data could be more appreciated.

In *Myf5^{Cre}:Rosa^{YFP}* mice, less than 10% of the satellite cells have never expressed *Myf5* and remain YFP negative. Therefore, the YFP negative population represents a rare subpopulation of satellite cells, evenly distributed across the myofibers. Over 40 fibres are counted per condition (>120 fibres per mouse) and due to the spatial distance (often >100 microns apart) between cell doublets, it is not possible to provide a single image that represents the entire sample. We chose instead to provide an example illustrating each type of YFP- divisions (asymmetric and symmetric).

15- Just FYI, cilia is plural and cilium is singular, thus the title should either be "by primary cilia" or "by the primary cilium".

We agree with the reviewer and we have modified the title as follows: "GLI3 processing by the primary cilium regulates muscle stem cell entry into G_{Alert} ".

16- Fig 4f-g: Please be consistent with the nomenclature for YFP vs. GFP (main text vs. figure legend vs. figure).

We corrected the nomenclature in **Supplementary Fig. 5g-j** and chose to use YFP for consistency.

17- Fig 7C: Please report the number of cells per condition. Also, why are there no error bars?

As requested by the reviewer, we have indicated the number of cells per conditions and included the exact number of cells in the **Data Source file**. We have also added the error bars.

References

- Addicks, GC, Brun, CE, Sincennes, MC, Saber, J, Porter, CJ, Francis Stewart, A, Ernst, P and Rudnicki, MA. **MLL1 is required for PAX7 expression and satellite cell self-renewal in mice.** *Nat Commun* 2019, **10**(1): 4256.
- Agarwal, NK, Qu, C, Kunkalla, K, Liu, Y and Vega, F. **Transcriptional regulation of serine/threonine protein kinase (AKT) genes by glioma-associated oncogene homolog 1.** *J Biol Chem* 2013, **288**(21): 15390-15401.
- Der Vartanian, A, Quetin, M, Michineau, S, Aurade, F, Hayashi, S, Dubois, C, Rocancourt, D, Drayton-Libotte, B, Szegedi, A, Buckingham, M, Conway, SJ, Gervais, M and Relaix, F. **PAX3 Confers Functional Heterogeneity in Skeletal Muscle Stem Cell Responses to Environmental Stress.** *Cell Stem Cell* 2019, **24**(6): 958-973 e959.
- Elia, D, Madhala, D, Ardon, E, Reshef, R and Halevy, O. **Sonic hedgehog promotes proliferation and differentiation of adult muscle cells: Involvement of MAPK/ERK and PI3K/Akt pathways.** *Biochim Biophys Acta* 2007, **1773**(9): 1438-1446.
- Fan, CW, Chen, B, Franco, I, Lu, J, Shi, H, Wei, S, Wang, C, Wu, X, Tang, W, Roth, MG, Williams, NS, Hirsch, E, Chen, C and Lum, L. **The Hedgehog pathway effector smoothened exhibits signaling competency in the absence of ciliary accumulation.** *Chem Biol* 2014, **21**(12): 1680-1689.
- Gustafsson, MK, Pan, H, Pinney, DF, Liu, Y, Lewandowski, A, Epstein, DJ and Emerson, CP, Jr. **Myf5 is a direct target of long-range Shh signaling and Gli regulation for muscle specification.** *Genes Dev* 2002, **16**(1): 114-126.
- Hammerschmidt, M, Bitgood, MJ and McMahon, AP. **Protein kinase A is a common negative regulator of Hedgehog signaling in the vertebrate embryo.** *Genes Dev* 1996, **10**(6): 647-658.
- Hardy, D, Besnard, A, Latil, M, Jouvion, G, Briand, D, Thepenier, C, Pascal, Q, Guguin, A, Gayraud-Morel, B, Cavaillon, JM, Tajbakhsh, S, Rocheteau, P and Chretien, F. **Comparative Study of Injury Models for Studying Muscle Regeneration in Mice.** *PLoS One* 2016, **11**(1): e0147198.
- Haycraft, CJ, Banizs, B, Aydin-Son, Y, Zhang, Q, Michaud, EJ and Yoder, BK. **Gli2 and Gli3 localize to cilia and require the intraflagellar transport protein polaris for processing and function.** *PLoS Genet* 2005, **1**(4): e53.
- Katoh, Y and Katoh, M. **Hedgehog target genes: mechanisms of carcinogenesis induced by aberrant hedgehog signaling activation.** *Curr Mol Med* 2009, **9**(7): 873-886.
- Keefe, AC, Lawson, JA, Flygare, SD, Fox, ZD, Colasanto, MP, Mathew, SJ, Yandell, M and Kardon, G. **Muscle stem cells contribute to myofibres in sedentary adult mice.** *Nat Commun* 2015, **6**: 7087.
- Klein, SD, Nguyen, DC, Bhakta, V, Wong, D, Chang, VY, Davidson, TB and Martinez-Agosto, JA. **Mutations in the sonic hedgehog pathway cause macrocephaly-associated conditions due to crosstalk to the PI3K/AKT/mTOR pathway.** *Am J Med Genet A* 2019, **179**(12): 2517-2531.
- Kopinke, D, Roberson, EC and Reiter, JF. **Ciliary Hedgehog Signaling Restricts Injury-Induced Adipogenesis.** *Cell* 2017, **170**(2): 340-351 e312.
- Laplante, M and Sabatini, DM. **mTOR signaling in growth control and disease.** *Cell* 2012, **149**(2): 274-293.
- Larsen, LJ and Moller, LB. **Crosstalk of Hedgehog and mTORC1 Pathways.** *Cells* 2020, **9**(10).
- Lee, G, Espirito Santo, AI, Zwingenberger, S, Cai, L, Vogl, T, Feldmann, M, Horwood, NJ, Chan, JK and Nanchahal, J. **Fully reduced HMGB1 accelerates the regeneration of multiple tissues by transitioning stem cells to GAlert.** *Proc Natl Acad Sci U S A* 2018, **115**(19): E4463-E4472.
- Li, J, Wang, C, Wu, C, Cao, T, Xu, G, Meng, Q and Wang, B. **PKA-mediated Gli2 and Gli3 phosphorylation is inhibited by Hedgehog signaling in cilia and reduced in Talpid3 mutant.** *Dev Biol* 2017, **429**(1): 147-157.
- Niewiadomski, P, Kong, JH, Ahrends, R, Ma, Y, Humke, EW, Khan, S, Teruel, MN, Novitch, BG and Rohatgi, R. **Gli protein activity is controlled by multisite phosphorylation in vertebrate Hedgehog signaling.** *Cell Rep* 2014, **6**(1): 168-181.
- Oprescu, SN, Yue, F, Qiu, J, Brito, LF and Kuang, S. **Temporal Dynamics and Heterogeneity of Cell Populations during Skeletal Muscle Regeneration.** *iScience* 2020, **23**(4): 100993.
- Pawlikowski, B, Pulliam, C, Betta, ND, Kardon, G and Olwin, BB. **Pervasive satellite cell contribution to uninjured adult muscle fibers.** *Skelet Muscle* 2015, **5**: 42.
- Piccioni, A, Gaetani, E, Palladino, M, Gatto, I, Smith, RC, Neri, V, Marcantoni, M, Giarretta, I, Silver, M, Straino, S, Capogrossi, M, Landolfi, R and Pola, R. **Sonic hedgehog gene therapy increases the ability of the dystrophic skeletal muscle to regenerate after injury.** *Gene Ther* 2014, **21**(4): 413-421.
- Renault, MA, Vandierdonck, S, Chapouly, C, Yu, Y, Qin, G, Metras, A, Couffinhal, T, Losordo, DW, Yao, Q, Reynaud, A, Jaspard-Vinassa, B, Belloc, I, Desgranges, C and Gadeau, AP. **Gli3 regulation of myogenesis is necessary for ischemia-induced angiogenesis.** *Circ Res* 2013, **113**(10): 1148-1158.
- Rion, N, Castets, P, Lin, S, Enderle, L, Reinhard, JR, Eickhorst, C and Ruegg, MA. **mTOR controls embryonic and adult myogenesis via mTORC1.** *Development* 2019, **146**(7).

- Rodgers, JT, King, KY, Brett, JO, Cromie, MJ, Charville, GW, Maguire, KK, Brunson, C, Mastey, N, Liu, L, Tsai, CR, Goodell, MA and Rando, TA. **mTORC1 controls the adaptive transition of quiescent stem cells from G0 to G(Alert).** *Nature* 2014, **510**(7505): 393-396.
- Rodgers, JT, Schroeder, MD, Ma, C and Rando, TA. **HGFA Is an Injury-Regulated Systemic Factor that Induces the Transition of Stem Cells into GAlert.** *Cell Rep* 2017, **19**(3): 479-486.
- Straface, G, Aprahamian, T, Flex, A, Gaetani, E, Biscetti, F, Smith, RC, Pecorini, G, Pola, E, Angelini, F, Stigliano, E, Castellot, JJ, Jr., Losordo, DW and Pola, R. **Sonic hedgehog regulates angiogenesis and myogenesis during post-natal skeletal muscle regeneration.** *J Cell Mol Med* 2009, **13**(8B): 2424-2435.
- Tiecke, E, Turner, R, Sanz-Ezquerro, JJ, Warner, A and Tickle, C. **Manipulations of PKA in chick limb development reveal roles in digit patterning including a positive role in Sonic Hedgehog signaling.** *Dev Biol* 2007, **305**(1): 312-324.
- Tukachinsky, H, Lopez, LV and Salic, A. **A mechanism for vertebrate Hedgehog signaling: recruitment to cilia and dissociation of SuFu-Gli protein complexes.** *J Cell Biol* 2010, **191**(2): 415-428.
- Tuson, M, He, M and Anderson, KV. **Protein kinase A acts at the basal body of the primary cilium to prevent Gli2 activation and ventralization of the mouse neural tube.** *Development* 2011, **138**(22): 4921-4930.
- Voronova, A, Coyne, E, Al Madhoun, A, Fair, JV, Bosiljcic, N, St-Louis, C, Li, G, Thurig, S, Wallace, VA, Wiper-Bergeron, N and Skerjanc, IS. **Hedgehog signaling regulates MyoD expression and activity.** *J Biol Chem* 2013, **288**(6): 4389-4404.
- Wen, X, Lai, CK, Evangelista, M, Hongo, JA, de Sauvage, FJ and Scales, SJ. **Kinetics of hedgehog-dependent full-length Gli3 accumulation in primary cilia and subsequent degradation.** *Mol Cell Biol* 2010, **30**(8): 1910-1922.
- Zeng, Q, Fu, Q, Wang, X, Zhao, Y, Liu, H, Li, Z and Li, F. **Protective Effects of Sonic Hedgehog Against Ischemia/Reperfusion Injury in Mouse Skeletal Muscle via AKT/mTOR/p70S6K Signaling.** *Cell Physiol Biochem* 2017, **43**(5): 1813-1828.

Reviewers' comments:

Reviewer #1 (Remarks to the Author):

The authors did a great job in addressing my previous concerns. The only suggestion I have is to remove "long-term" in the second last sentence of the abstract: "As a result, satellite cells lacking GLI3 display rapid cell-cycle entry, increased proliferation and augmented self renewal, and markedly enhanced long-term regenerative capacity." Not only is the long-term regeneration not examined in the study, the statement is also inconsistent with the message conveyed in the second last paragraph in the discussion, where the authors try to speculate a long-term defects of the Gli3 KOs. In fact, there is evidence supporting that satellite cells maintained at a Galert state due to loss of Pten leads to regenerative deficiencies in the long-term (PMID: 27880908). Inclusion of this reference in the discussion may strengthen the authors argument.

Reviewer #2 (Remarks to the Author):

The revision address several of the criticisms raised during the first round of reviews. However, a fundamental problem remains, and that is the absence of data supporting the proposition that HH signaling is regulated by cilia-associated Gli and Gli processing in what the authors refer to as QSCs. The authors add experiments to alter the processing using knockdown of Gli3 or Ift88, but all of these experiments are done in myoblasts (Supplemental Figure 2), so they have no relevance to QSCs. Furthermore, as per the authors' own data, myoblasts are devoid of cilia, so the interpretation of these results does not support the general conclusions that the authors draw. This major conclusion is further undermined by the additional transcriptional studies (Figure 4) that fail to demonstrate any differences in expression of Gli targets between wild-type and Gli-deficient SCs, again undermining the contention that the HH signaling pathway is regulated by cilia-associated Gli in QSCs. And the one QSC phenotype in the Gli KO in QSCs, the G0-to-Galert transition, is mediated by mTORC1 signaling, but no mechanistic studies are done to connect Gli processing to mTORC1 signaling in QSCs.

The lack of these primary mechanistic studies were the main limitation of the initial submission and they remain here. It seems likely that the regeneration changes reported in Figure 8 are likely due to the pleiotropic effects of Gli deletion in the proliferating myoblasts and not in QSCs. This is highlighted by the fact Gli deletion in QSCs does not lead to any increase in QSC activation until 4 weeks after deletion (Fig. 3b), a delay that is difficult to understand in terms of the idea of Gli being important for maintenance of SCs as quiescent stem cells.

Reviewer #3 (Remarks to the Author):

I applaud the authors for investing a lot of resources and hard work in submitting a markedly improved manuscript. I especially appreciate the disclosure of all the primary data and increasing the number of biological replicates, which did indeed change some of the initial conclusions. However, now that the manuscript is much more complete and easier to digest, there is one major conceptual issue, which still requires clarification. The authors describe

that there is a difference in the ciliation frequency between QSCs, which are ciliated, and myoblasts, which are unciliated. The authors further note that loss of Gli3 leads to Hh de-repression in myoblasts but not in QSCs and propose a model, without any functional data, that Gli3 in QSCs controls their quiescence via the mTORC pathway. Because of the difference in ciliation between the two models used here, it raises concerns about the interpretation of the results (Hh de-repression in only one model vs. mTORC in both?!). In addition, as this is a novel proposed mechanism on how cilia control quiescence via Gli3-mTORC (which would be really fascinating), some functional evidence is required.

Reviewers' comments:

We thank the reviewers for their global appreciation of our revised manuscript. Taking into consideration their comments, we have extensively modified the manuscript in order to clarify our take-home message that the Hedgehog mediator GLI3 regulates muscle stem cell function through a mTORC1 signaling-dependent mechanism. Following the reviewers' suggestions, we performed additional experiments to strengthen our results and we hope that the new version of our manuscript will satisfy their expectations.

Reviewer #1 (Remarks to the Author):

The authors did a great job in addressing my previous concerns. The only suggestion I have is to remove "long-term" in the second last sentence of the abstract: "As a result, satellite cells lacking GLI3 display rapid cell-cycle entry, increased proliferation and augmented self-renewal, and markedly enhanced long-term regenerative capacity." Not only is the long-term regeneration not examined in the study, the statement is also inconsistent with the message conveyed in the second last paragraph in the discussion, where the authors try to speculate a long-term defects of the Gli3 KOs. In fact, there is evidence supporting that satellite cells maintained at a Galert state due to loss of Pten leads to regenerative deficiencies in the long-term (PMID: 27880908). Inclusion of this reference in the discussion may strengthen the authors' argument.

We thank the reviewer for his positive feedback on our work. We agree that the word 'long-term' is not appropriated and we removed it accordingly. As well, we have now cited the reference PMID: 27880908 regarding the role of Pten on GALert, as indeed it further strengthens our conclusion.

Reviewer #2 (Remarks to the Author):

The revision address several of the criticisms raised during the first round of reviews. However, a fundamental problem remains, and that is the absence of data supporting the proposition that HH signaling is regulated by cilia-associated Gli and Gli processing in what the authors refer to as QSCs. The authors add experiments to alter the processing using knockdown of Gli3 or Ift88, but all of these experiments are done in myoblasts (Supplemental Figure 2), so they have no relevance to QSCs. Furthermore, as per the authors' own data, myoblasts are devoid of cilia, so the interpretation of these results does not support the general conclusions that the authors draw. This major conclusion is further undermined by the additional transcriptional studies (Figure 4) that fail to demonstrate any differences in expression of Gli targets between wild-type and Gli-deficient SCs, again undermining the contention that the HH signaling pathway is regulated by cilia-associated Gli in QSCs. And the one QSC phenotype in the Gli KO in QSCs, the G0-to-Galert transition, is mediated by mTORC1 signaling, but no mechanistic studies are done to connect Gli processing to mTORC1 signaling in QSCs.

We respectfully disagree with this reviewer on his claim that myoblasts are "devoid of ciliation". Our data clearly demonstrates that myoblasts are indeed ciliated, but to a lesser extent than observed in quiescent satellite cells because they are continuously cycling.

Most of the cell types build a primary cilium upon cell cycle exit, in G₀^{1,2}. Thus, it is not surprising that the non-cycling, quiescent satellite cells exhibit and maintain a primary cilium (**Fig. 1; Supplementary Fig. 2**)^{3,4}. The maintenance of satellite cell quiescence within the homeostatic muscle allows for an easy visualization and quantification of primary cilia at their surface. However, proliferating cells, such as myoblasts, are asynchronously, transiently and dynamically ciliated, exhibiting mainly a primary cilium in G1 and S phases (**Fig. 1**)⁵⁻⁷. Moreover, primary cilia length decreases sharply as cells enter the cell cycle

^{5,6}. Therefore, only $\alpha\text{TUB}^+/\text{ARL13B}^+$ elongated structures ($> 0.5\mu\text{m}$) were enumerated to avoid counting ciliary vesicles or centrosomes as cilia (**Fig. 1, 2; Supplementary Fig. 2**). We have now specified it in the ‘results’ section of our manuscript to improve clarity of our data and interpretation.

Although myoblasts dynamically rearrange primary cilia, they can transduce the Hedgehog signaling similarly to ‘stably’ ciliated cells ^{8,9}. Knocking-down *Gli3* (*siGli3*, **Supplementary Fig. 2h-m**), blocking ciliation (*silft88*, **Supplementary Fig. 2h-l**) or knocking-out *Gli3* (*Gli3^{Δ/Δ}*, **Supplementary Fig. 6c**) lift the GLI3R-induced inhibition on canonical Hedgehog downstream target genes in ciliated myoblasts. Similarly, knocking-out *Gli3* in quiescent (*Gli3^{Δ/Δ}* QSCs) and activated satellite cells (*Gli3^{Δ/Δ}* ASCs) increases slightly the expression of the GLI-target genes (**Supplementary Figs. 5b, c; 6a, b**). The minor effects on GLI3 canonical downstream target genes is likely due to the heterogeneity of our FACS-sorted cell populations that undermines potential variations in the expression of GLI-target genes (**Fig. 1; Supplementary Fig. 1a, b**). Indeed, QSCs are partially activated ¹⁰⁻¹² and ASCs contain both proliferating SCs and myogenic progenitors ¹³, which all display high variations in Hedgehog-related gene expression. We have now clarified this point in our ‘results’ and ‘discussion’ sections.

However, canonical Hedgehog signaling cannot be solely responsible for the *Gli3^{Δ/Δ}* phenotype, since the expression of its canonical downstream target genes are mainly turned off as QSCs activate. Instead, we propose that mTORC1 activation upon GLI3R depletion is responsible for the enhanced muscle regeneration observed in *Gli3^{Δ/Δ}* mice ¹⁴⁻¹⁷. As requested by reviewer #3 (see response below), we have now performed functional experiments to link GLI3R function and mTORC1 signaling in muscle cells (**Fig. 7; Supplementary Fig. 7**). Consistent with our RNA-sequencing data, we showed that loss of the Hedgehog mediator GLI3R leads to a slight upregulation of the expression of canonical Hedgehog downstream target genes and triggers mTORC1 signaling activation.

Thus, our data highlight the existence of a crosstalk between ciliary GLI3-mediated Hedgehog and mTORC1 signaling to regulate satellite cell and myogenic progenitor function.

The lack of these primary mechanistic studies were the main limitation of the initial submission and they remain here. It seems likely that the regeneration changes reported in Figure 8 are likely due to the pleiotropic effects of Gli deletion in the proliferating myoblasts and not in QSCs.

Indeed, GLI3 has pleiotropic effects *in vivo*, making them difficult to decipher precisely. To overcome this limitation, we performed several analysis on FACS-sorted satellite cells to demonstrate that loss of GLI3 triggers the G_0 -to- G_{Alert} transition in QSCs (**Fig. 5; Supplementary Fig. 5**) and increases the proliferation capacity of ASCs (**Fig. 6; Supplementary Fig. 6**).

Consistent with these results, *ex vivo* satellite cell analysis on cultured myofibers revealed increased ability of *Gli3^{Δ/Δ}* QSCs to enter the cell cycle (**Fig. 4e, f**) and increased EdU incorporation of *Gli3^{Δ/Δ}* ASCs (**Figs. 4g, h; 5h; 6c, d**). Using myofiber culture and satellite cell-derived myoblasts, we showed that loss of GLI3 delays but does not prevent differentiation (**Figs. 4i, j; 5e**). All these data were further strengthened by the transcriptomic analysis performed on QSCs and ASCs (**Figs. 5, 6**). Therefore and as already mentioned in the discussion of our manuscript, the enhanced ability to regenerate of the *Gli3^{Δ/Δ}* mice results from the pleiotropic effects of GLI3 in satellite cells.

This is highlighted by the fact Gli deletion in QSCs does not lead to any increase in QSC activation until 4 weeks after deletion (Fig. 3b), a delay that is difficult to understand in terms of the idea of Gli being important for maintenance of SCs as quiescent stem cells.

We observed a slight increase in the number of satellite cells post-tamoxifen treatment in the *Gli3^{Δ/Δ}* mice (**Supplementary Fig. 3d, e**). However, this number remains stable over time, and our results indicate that *Gli3^{Δ/Δ}* satellite cells return to and maintain quiescence under homeostatic conditions (**Supplementary**

Fig. 3f, g). Satellite cells contribute to uninjured myofiber homeostasis in adulthood^{18,19}. Since *Gli3* deletion leads to G_{Alert} , enhanced cell proliferation and self-renewal, we speculate that the initial depletion of *Gli3* triggers a transient entry into the cell cycle with a preference for self-renewal, increasing the total number of satellite cells before homeostatic quiescence is re-asserted. Of note, GLI3R is essential to maintain G_0 quiescence as *Gli3* ^{Δ/Δ} satellite cells transition from G_0 to G_{Alert} within 3 days post-tamoxifen treatment in resting muscle (**Fig. 5, 7**).

Reviewer #3 (Remarks to the Author):

I applaud the authors for investing a lot of resources and hard work in submitting a markedly improved manuscript. I especially appreciate the disclosure of all the primary data and increasing the number of biological replicates, which did indeed change some of the initial conclusions. However, now that the manuscript is much more complete and easier to digest, there is one major conceptual issue, which still requires clarification. The authors describe that there is a difference in the ciliation frequency between QSCs, which are ciliated, and myoblasts, which are unciliated. The authors further note that loss of *Gli3* leads to Hh de-repression in myoblasts but not in QSCs and propose a model, without any functional data, that *Gli3* in QSCs controls their quiescence via the mTORC pathway. Because of the difference in ciliation between the two models used here, it raises concerns about the interpretation of the results (Hh de-repression in only one model vs. mTORC in both?!). In addition, as this is a novel proposed mechanism on how cilia control quiescence via *Gli3*-mTORC (which would be really fascinating), some functional evidence is required.

We thank the reviewer for his overall appreciation of our revised manuscript. We apologize for the confusion brought by our description of ciliation in QSCs and myoblasts. As mentioned above (response to reviewer 2), we have now clarified this point.

To further support the existence of a crosstalk between GLI3-mediated Hedgehog and mTORC1 signaling that regulates muscle stem cell function, we have now provided functional evidence using *in vivo* and *ex vivo* experiments (**Fig. 7 and Supplementary Fig. 7**). In brief, we showed that FSK-mediated GLI3R processing decreases mTORC1 activation in *Gli3* ^{$+/+$} cells, which in turn, inhibits *Gli3* ^{$+/+$} satellite cell activation and myoblast proliferation (**Supplementary Fig. 6 and Supplementary Fig. 8c**). However, FSK treatment in *Gli3* ^{Δ/Δ} cells does not affect their ability to activate and proliferate faster (**Fig. 7e-h; Supplementary Fig. 7a-e**). This confirms that GLI3 proteolytic processing upon FSK mediated-PKA activation regulates mTORC1 signaling. In parallel, Rapamycin-induced mTORC1 inhibition restores G_0 quiescence features in *Gli3* ^{Δ/Δ} QSCs to the ones observed in *Gli3* ^{$+/+$} QSCs (**Fig. 7a-d**). Moreover, Rapamycin blocks the activation and proliferation of muscle cells in both *Gli3* ^{$+/+$} and *Gli3* ^{Δ/Δ} mice. Therefore, these new data demonstrate that the Hedgehog mediator GLI3R acts upstream mTORC1 signaling to regulate muscle stem cell and myogenic progenitor function.

- 1 Sanchez, I. & Dynlacht, B. D. Cilium assembly and disassembly. *Nature cell biology* **18**, 711-717, doi:10.1038/ncb3370 (2016).
- 2 Satir, P., Pedersen, L. B. & Christensen, S. T. The primary cilium at a glance. *Journal of cell science* **123**, 499-503, doi:10.1242/jcs.050377 (2010).
- 3 Jaafar Marican, N. H., Cruz-Migoni, S. B. & Borycki, A. G. Asymmetric Distribution of Primary Cilia Allocates Satellite Cells for Self-Renewal. *Stem cell reports*, doi:10.1016/j.stemcr.2016.04.004 (2016).
- 4 Venugopal, N. *et al.* The primary cilium dampens proliferative signaling and represses a G2/M transcriptional network in quiescent myoblasts. *BMC molecular and cell biology* **21**, 25, doi:10.1186/s12860-020-00266-1 (2020).
- 5 Pugacheva, E. N., Jablonski, S. A., Hartman, T. R., Henske, E. P. & Golemis, E. A. HEF1-dependent Aurora A activation induces disassembly of the primary cilium. *Cell* **129**, 1351-1363, doi:10.1016/j.cell.2007.04.035 (2007).
- 6 Phua, S. C. *et al.* Dynamic Remodeling of Membrane Composition Drives Cell Cycle through Primary Cilia Excision. *Cell* **168**, 264-279 e215, doi:10.1016/j.cell.2016.12.032 (2017).
- 7 Ford, M. J. *et al.* A Cell/Cilia Cycle Biosensor for Single-Cell Kinetics Reveals Persistence of Cilia after G1/S Transition Is a General Property in Cells and Mice. *Developmental cell* **47**, 509-523 e505, doi:10.1016/j.devcel.2018.10.027 (2018).
- 8 Ho, E. K., Tsai, A. E. & Stearns, T. Transient Primary Cilia Mediate Robust Hedgehog Pathway-Dependent Cell Cycle Control. *Current biology : CB* **30**, 2829-2835 e2825, doi:10.1016/j.cub.2020.05.004 (2020).
- 9 Barzi, M., Berenguer, J., Menendez, A., Alvarez-Rodriguez, R. & Pons, S. Sonic-hedgehog-mediated proliferation requires the localization of PKA to the cilium base. *Journal of cell science* **123**, 62-69, doi:10.1242/jcs.060020 (2010).
- 10 van Velthoven, C. T. J., de Morree, A., Egner, I. M., Brett, J. O. & Rando, T. A. Transcriptional Profiling of Quiescent Muscle Stem Cells In Vivo. *Cell reports* **21**, 1994-2004, doi:10.1016/j.celrep.2017.10.037 (2017).
- 11 Machado, L. *et al.* In Situ Fixation Redefines Quiescence and Early Activation of Skeletal Muscle Stem Cells. *Cell reports* **21**, 1982-1993, doi:10.1016/j.celrep.2017.10.080 (2017).
- 12 Yue, L., Wan, R., Luan, S., Zeng, W. & Cheung, T. H. Dek Modulates Global Intron Retention during Muscle Stem Cells Quiescence Exit. *Developmental cell* **53**, 661-676 e666, doi:10.1016/j.devcel.2020.05.006 (2020).
- 13 Oprescu, S. N., Yue, F., Qiu, J., Brito, L. F. & Kuang, S. Temporal Dynamics and Heterogeneity of Cell Populations during Skeletal Muscle Regeneration. *iScience* **23**, 100993, doi:10.1016/j.isci.2020.100993 (2020).
- 14 Rion, N. *et al.* mTOR controls embryonic and adult myogenesis via mTORC1. *Development* **146**, doi:10.1242/dev.172460 (2019).
- 15 Matsumoto, A. *et al.* mTORC1 and muscle regeneration are regulated by the LINC00961-encoded SPAR polypeptide. *Nature* **541**, 228-232, doi:10.1038/nature21034 (2017).
- 16 Zhang, P. *et al.* mTOR is necessary for proper satellite cell activity and skeletal muscle regeneration. *Biochemical and biophysical research communications* **463**, 102-108, doi:10.1016/j.bbrc.2015.05.032 (2015).
- 17 Rodgers, J. T. *et al.* mTORC1 controls the adaptive transition of quiescent stem cells from G0 to G(Alert). *Nature* **510**, 393-396, doi:10.1038/nature13255 (2014).
- 18 Pawlikowski, B., Pulliam, C., Betta, N. D., Kardon, G. & Olwin, B. B. Pervasive satellite cell contribution to uninjured adult muscle fibers. *Skeletal muscle* **5**, 42, doi:10.1186/s13395-015-0067-1 (2015).
- 19 Keefe, A. C. *et al.* Muscle stem cells contribute to myofibres in sedentary adult mice. *Nature communications* **6**, 7087, doi:10.1038/ncomms8087 (2015).

REVIEWERS' COMMENTS

Reviewer #3 (Remarks to the Author):

The authors have addressed all my concerns. Congrats!

Reviewer #4 (Remarks to the Author):

I believe Reviewer #2 makes some valid points. Either additional experiments should be performed or the data described appropriately. Moreover, a non-canonical non ciliary signaling pathway should be considered.

Brun et al. show a new role for Gli3 in muscle satellite cells (SCs) and regeneration. Using a SC specific conditional knockout model for Gli3, they show loss of Gli3, potentially due to its repressor form, leads to an activated state of SCs which promotes increased regeneration kinetics. Authors propose a mechanism of Gli3 promoting mTORC1 signaling which regulates the activation status of SCs. Authors show a role for Gli3 in SCs, but many of the conclusions are overstated given the data presented.

Problems:

- The “reserve” cells used are a poor surrogate for muscle stem cells – they were defined in the 1998 (ref 31) prior to the identification of markers for isolating stem cells by FACS. They are a population of starved myoblasts (committed progenitors) – not quiescent stem cells. That they possess cilia is unexpected but they are not bona fide stem cells. To claim they are SCs, the function of “reserve” cells in regeneration assays should be compared with SCs and myoblasts in transplantation studies to validate this surrogate cell type.
- Additionally it is strange that Gli targets are not seen in the transcriptome of wild-type compared to Gli-deficient satellite cells – raising questions about the HH signaling pathway. In other words, the authors claim a major role for Gli3 in ciliary pathways, but no ciliary targets are identified in the RNAseq.
- I also agree with the reviewer that the connection between Gli signaling and mTORC1 signaling is missing, and that link is important to the thesis of the paper. Based on their data, the authors should consider non-canonical signaling, independent of cilia, from Gli3 could be occurring and should analyze this possibility. Accordingly, to understand if Gli3 has crosstalk with other Gli TFs, an antagonist of Gli1/2 such as GANT61 could be tested. Similarly, Smo or Ptch1 should be considered to understand if these have a role in Gli3 signaling in vitro.
- A point not raised by the reviewer is that data are lacking to establish the roles of the isoforms with opposing roles – activating Gli3-FL and cleaved Gli3-repressor form – This would require loss of function(knockdown) and gain of function(overexpression) studies in satellite cells, which although difficult, can be done and has been done by others for other gene products.
- Confirmation of lack of Hh in conditioned medium of myoblasts, or myofibers should be performed. An effect of Hh cannot be ruled out if it is present in media due to secretion by the cells.

There are definitely merits to the manuscript but If accepted, the authors should be required modify their claims to those that are substantiated and change the language accordingly.

P3. "Expression analyses on QSCs, ASCs, proliferating myoblasts and differentiated myotubes further confirmed that only Gli3 is enriched in ASCs and proliferating myoblasts (Fig. 1a), supporting the hypothesis that only GLI3 participates in Hedgehog signal transduction across myogenic progression" ◇ There is not sufficient evidence in the manuscript to state Gli3 is the only Hh signaling important for myogenesis. This is actually clear from their own studies showing no differences in ciliary pathways when Gli3 KO is performed in SCs.

P6. "Collectively, these results show that GLI3 is mainly processed to a repressor in QSCs."
◇ An analysis of the different Gli3 isoforms in QSCs and ASCs is lacking and necessary to make this statement. The analysis was performed in myoblasts and not QSCs or ASCs.

P7. "Thus, our results demonstrate that the ratio of GLI3FL/GLI3R relies primarily on cilia dynamics and likely not exogenous Hedgehog signals, as satellite cells progress through the myogenic lineage and self-renew." ◇ There is not sufficient information in the current manuscript to support this claim. No analysis of Hedgehog signaling is performed in vivo. Cells grown in monolayer with defined medium rich in serum with multiple growth factors do not resemble the in vivo Hh signaling microenvironment. Note that Hh could be present in the cultures as it could be secreted by myoblasts.